# Transposable elements maintain genome-wide heterozygosity in inbred populations

Hanne De Kort [1] ✉, Sylvain Legrand[2], Olivier Honnay[1] & James Buckley[3]

Elevated levels of inbreeding increase the risk of inbreeding depression and extinction, yet many inbred species are widespread, suggesting that inbreeding has little impact on evolutionary potential. Here, we explore the potential for transposable elements (TEs) to maintain genetic variation in functional genomic regions under extreme inbreeding. Capitalizing on the mixed mating system of *Arabidopsis lyrata*, we assess genome-wide heterozygosity and signatures of selection at single nucleotide polymorphisms near transposable elements across an inbreeding gradient. Under intense inbreeding, we find systematically elevated heterozygosity downstream of several TE superfamilies, associated with signatures of balancing selection. In addition, we demonstrate increased heterozygosity in stress-responsive genes that consistently occur downstream of TEs. We finally reveal that TE superfamilies are associated with specific signatures of selection that are reproducible across independent evolutionary lineages of *A. lyrata*. Together, our study provides an important hypothesis for the success of self-fertilizing species.

Inbreeding has been consistently shown to reduce individual fitness, population viability and evolutionary potential[1–3], yet many inbred species still persist, and even thrive, in a broad range of environments[4,5]. This evolutionary paradox is exemplified by the success of self-pollinating plant species, which are particularly capable of colonizing a wide range of environments[6]. *Arabidopsis thaliana*, for instance, a self-compatible species with extreme inbreeding levels across its natural range, does not show widespread signatures of increased extinction risk but instead exhibits significant adaptive potential[7–9]. Thus, while small and genetically depleted populations frequently continue to evolve and persist, we are still far from understanding the molecular mechanisms underlying this evolutionary potential.

Insufficient rates of outcrossing can cause populations to suffer inbreeding depression due to recessive deleterious mutations becoming homozygous, consequently increasing population extinction risk[10–13]. Empirical work has shown that populations with large effective sizes in particular can be vulnerable to inbreeding depression, because a long history of outcrossing has allowed the accumulation of deleterious mutations that become visible to negative selection when inbreeding levels increase[2,13]. Small and/or self-compatible populations, on the other hand, more effectively purge deleterious genetic load, allowing them to maintain fitness during episodes of high inbreeding[2,14–17], but see ref. 18. Purging of alleles associated with inbreeding depression therefore is thought to contribute to the success of self-fertilizing species[16,19].

While purging can safeguard populations from inbreeding depression, it may facilitate the depletion of population genetic variation under prolonged inbreeding. When low recombination rates prevent the segregation of neutral and adaptive genetic variation from recessive deleterious alleles (cfr. Hill-Robertson interference[20–22]), purging may particularly rapidly remove genetic variation required for long-term evolution and fitness. In addition, slightly deleterious alleles can become adaptive in specific environmental conditions[23–25], or in combination with compensatory mutations[26], causing purging to remove cryptic evolutionary potential associated with conditionally deleterious alleles. While inbreeding and purging may thus mutually accelerate the loss of evolutionary potential, self-compatible species frequently appear to be relatively resilient to high levels of inbreeding[27–32].

[1]Plant Conservation and Population Biology, University of Leuven, Kasteelpark Arenberg 31-2435, BE-3001 Leuven, Belgium. [2]Univ. Lille, CNRS, UMR 8198 - Evo-Eco-Paleo, F-59000 Lille, France. [3]School of Biological and Marine Sciences, University of Plymouth, Plymouth PL1 2BT, UK. ✉e-mail: hanne.dekort@kuleuven.be

A poorly explored mechanism that may govern evolution in naturally inbred species involves mutation rate variability across the genome. Transposable elements (TEs) can rapidly generate genetic variation through increasing mutation rates of upstream and downstream flanking genomic regions;[33–36] several studies have observed increases over 10x the background mutation rate[36,37]. Although such mutations can be lethal, causing natural selection to systematically purge TEs and their hosts from the gene pool[38], many TEs have been co-opted by the host to adjust gene expression in response to local environmental conditions[34,39–41]. As TEs can impact mutation rates up to 3.5 kb away[37,42], much of the genome may be under the direct influence of transposable elements.

Two mechanisms can generate genetic diversity near TEs. First, TE excision and insertion during transposition events require annealing and repairing of flanking DNA by DNA repair machinery[34,35,37]. Because DNA repair is an error-prone process, this mutation-by-transposition mechanism is considered to be a major source of mutations near TEs[37]. Second, TEs and nearby sequences tend to be heavily methylated. This methylated DNA, cytosines in particular, is susceptible to spontaneous mutation, resulting in increased cytosine-to-thymine conversion[33,36,43–45]. As a consequence, even silenced TEs may impact local mutation rates, and the mere presence of TEs may generate functional genetic diversity where host genomes have co-opted TEs[40,46–48]. The efficiency of this mutation-by-methylation mechanism may vary across TEs depending on their activity in the host genome. Stress-responsive TEs, including at least several Copia elements, frequently contain genetic elements conferring downstream stress-responsiveness through activation (de-methylation) or repression (methylation) under stress[38,49–51]. Stress-responsive TEs are thus characterized by prolonged periods of downstream methylation during or between stress episodes, which may increase mutation rates and genetic diversity particularly downstream of these TEs and in functional genes. Similarly, miniature inverted-repeat transposable elements (MITEs) have been shown to attract siRNAs that repress downstream gene expression[52], and to impact methylation of a downstream flowering gene in maize[53]. Whether genomic regions downstream of specific TE families are systematically characterized by increased genetic diversity and involvement in evolution remains to be elucidated.

Co-opted TEs may increase mutation rates particularly in functional genomic regions that can benefit from genetic variation. They have correspondingly been shown to be non-randomly distributed across the genome, preferentially integrating within functional genes and away from vital core functions[40,41,54,55]. Experimental work has also shown that, upon TE activation, transposition can exponentially increase mutation rates in genes without harmful effects on fitness[55]. This suggests that enhanced TE activity has the potential to reinforce evolution through generating neutral and/or beneficial genetic variation in functional genes. Recent findings of high genetic diversity in TE-rich genomic regions of invasive ant populations[56] may point to a role for genetic bottlenecks and inbreeding in driving TE activity and nearby genetic diversity. In the mixed mating species *A. lyrata*, self-fertilizing populations are characterized by increased TE copy numbers and higher TE frequencies than outcrossing populations, aligning with models of neutral evolution and ectopic recombination[57]. More specifically, the higher levels of sequence homozygosity in self-fertilizing populations reduces the rate at which heterologous sequences recombine, thereby generating harmful chromosomal rearrangements. As a consequence, self-fertilization can facilitate genome-wide TE accumulation, potentially providing considerable opportunities for evolution in genomic sequences flanking TEs. However, while inbreeding may affect TE dynamics and evolution in *A. lyrata*[57–59], the impact of inbreeding on genetic variation and evolution surrounding TEs has never been explicitly tested.

Here, we analyze the relationship between TEs and the probability of heterozygosity at nearby loci across an inbreeding gradient in North American *A. lyrata*, where populations exhibit a mixed mating system[60,61]. In addition, we assess the proportion of flanking adaptive outliers per TE superfamily and validate these TE-related evolutionary signatures in populations of an independent evolutionary lineage (European *A. lyrata*). We finally apply gene ontology enrichment analyses to test whether genetic diversity near TEs is associated with particular functions (e.g. related to stress responses). We hypothesize that (i) TEs can impact heterozygosity across the genome, particularly downstream, (ii) this increase in heterozygosity is functional and may thus provide a basis for evolution under inbreeding, and (iii) the role of TEs in evolution is consistent across sampling areas (Fig. 1). Because TEs are thought to integrate near functional gene regions but away from vital core functions, we further hypothesize that genetic variation downstream of TEs arises in redundant genes that are not essential to organismal functioning. We show that transposable elements help shaping the genome-wide heterozygosity landscape in inbred populations.

## Results

### Genetic variation near TEs

In order to understand the relationship between TEs and nearby heterozygosity, we first explored the tendency of random SNPs (Supplementary Data 1–4) obtained through restriction-site associated DNA sequences (RADseq) to occur near TEs. We focused on 48 individuals of *A. lyrata* varying in the degree of inbreeding and mating system (see Supplementary Data 5 for origin and mating characteristics). We found that a SNP was on average 3467 bp away from a TE, and the genomic distance between each SNP and its nearest TE varied between 0 bp (16 SNPs were within a TE) and 27,275 bp (Fig. 2). SNPs were most frequently located near MuDR TEs (292 SNPs), and least frequently near Tase TEs (23 SNPs; Fig. 2). The MuDR superfamily is featured by an intermediate abundance of TEs in *A. lyrata*[54], but is considered highly mutagenic[47], suggesting that TE superfamily characteristics rather than TE abundance determines nearby SNP abundance. On average, SNPs were closest to TEs from the CACTA superfamily (3123 bp, ranging from 73 to 13,896 bp) and most distal to TEs from the Harbinger superfamily (4595 bp, ranging from 65 to 21,559 bp).

As expected under high levels of inbreeding ($F_{IS} > 0.6$), heterozygosity was low across most of the genome (Fig. 3a). However, Copia, Harbinger and LINE elements manifested elevated levels of nearby heterozygosity under inbreeding ($F = 4.98**$ based on generalized mixed modeling, Supplementary Data 6), particularly downstream (Fig. 3a, $F = 4.36**$, Supplementary Data 6). We refer to $H_{TE}$ as heterozygosity that remains high under inbreeding due to the presence of nearby TEs (here Copia and Harbinger elements). While Copia and Harbinger elements were associated with elevated levels of heterozygosity particularly under inbreeding, LINE elements had high heterozygosity irrespective of inbreeding levels (Fig. 3b). The TE Effect thus was limited to Copia and Harbinger elements. Heterozygosity also increased with increasing genomic distance from TEs, up to ca. 3.5 kb, after which it generally decreased again (Fig. 2c), but this effect was not significant (Supplementary Data 6). Across TE superfamilies, heterozygosity was significantly higher downstream than upstream of TEs (Fig. 3a, b and Supplementary Data 6). Together, these findings suggest that TEs can impact downstream heterozygosity across the genome, in accordance with our hypothesis (Fig. 1-Q1). While Copia and Harbinger elements were associated with increased levels of nearby heterozygosity under inbreeding ($F_{IS} > 0.6$), average heterozygosity levels (across inbreeding coefficients) did not differ from those near other TE superfamilies (Fig. 3c vs. d).

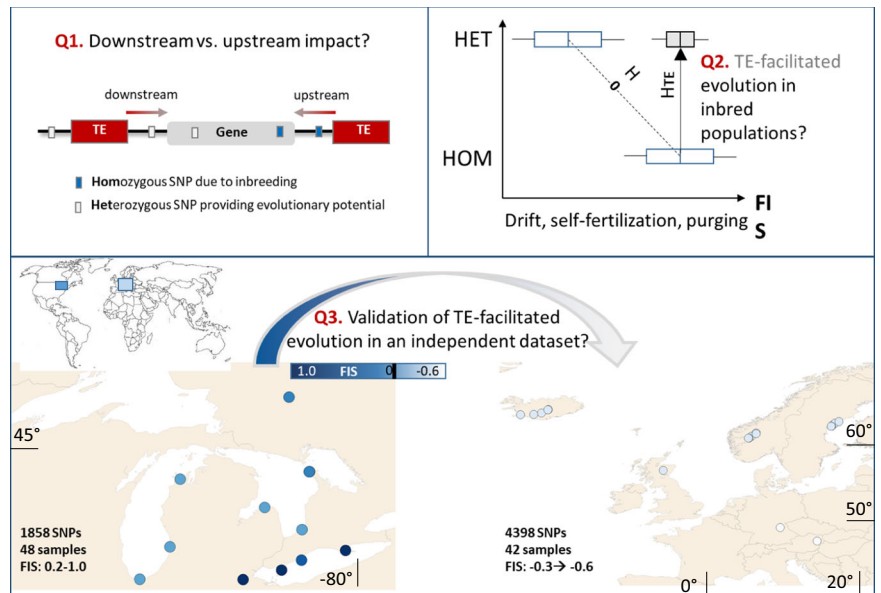

**Fig. 1 | Overview of research questions.** Research questions (Q1-Q3) on the role of TE-facilitated evolution under inbreeding, with $F_{IS}$ representing the genome-wide inbreeding coefficient. In Q2, the gray boxplot represents a TE-mediated shift in the expected negative relationship between inbreeding and heterozygosity from $H_0$ to $H_{TE}$. For Q3, the left panel shows the North American Great Lakes region (left) where there is significant variation in inbreeding in *A. lyrata* ssp. *lyrata* and the right panel shows Northern European populations of *A. lyrata* ssp. *petraea*. The blue rectangles on the world map represent the study locations (Great Lakes in dark blue and Europe in light blue). The arrow in the lower panel (Q3) hypothesizes extrapolation of the findings observed for the American lineage towards the European lineage.

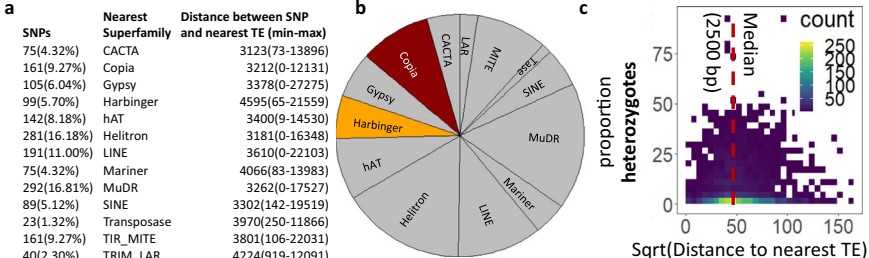

**Fig. 2 | Overall relationship between random SNPs and nearest TE.** Panels represent TE superfamily properties, including number of SNPs near TE superfamilies (**a**, **b**), and distance between SNPs and their nearest TE (**c**). Panels are based on 1734 independent SNPs. Source Data are provided as a Source Data file.

## Signatures of selection near TEs

Based on genomic outlier detection methods, we found that genome-wide signatures of natural selection (Fig. 4a and Supplementary Data 7) were similar upstream and downstream of TEs (Figs. 3d and 4b), but varied considerably across TE superfamilies (Fig. 4c). While some superfamilies were characterized by elevated levels of balancing selection (e.g. downstream of Copia and Harbinger, Figs. 3c and 4c), others manifested pronounced levels of divergent selection (e.g. upstream of Gypsy and near Helitron, Fig. 4c). The SNPs showing signatures of balancing selection and SNPs downstream of TEs tended to be heterozygous (Figs. 3a and 4d, e), and they also resided in single copy genes (Supplementary Fig. 1), indicating that they were more essential to organismal functioning than SNPs upstream of TEs and without signatures of selection. Because signatures of purifying selection can resemble those of balancing selection, we also explored the behavior of non-synonymous and synonymous substitutions in our set of putative balancing SNPs. Specifically, purifying selection more readily constrains non-synonymous substitutions to low frequency than balancing selection, and can therefore be identified as low-frequency signatures of purifying selection. While SNPs near CACTA elements appeared to be associated with low minor allele frequencies, which could point to purifying selectin acting on deleterious alleles, we found signatures of purifying selection to be generally rare, particularly in the North American lineage (Supplementary Fig. 2). Genomic regions with signatures of balancing selection were also significantly enriched for processes that have been frequently associated with balancing selection (e.g. response to biotic stimuli; Supplementary Data 8), suggesting that our findings are not strongly impacted by purifying selection.

Although regions downstream of TEs did not show elevated levels of natural selection as compared to regions upstream of TEs (Fig. 4b, $F = 3.35$, Supplementary Data 6), they were notably associated with stress response functions (Fig. 4f). Specifically, a total of 14 out of 17 SNPs associated with responses to abiotic stimulus were downstream of TEs, as well as 9 out of 11 SNPs associated with response to biotic stimulus (Fig. 4f and Supplementary Data 8). Interestingly, SNPs downstream of (i) MuDR TEs were enriched for processes involved in photosynthesis and growth, (ii) Helitron were typically associated with abiotic stresses (heat, light, drought), and (iii) MITEs with wounding and disease (Supplementary Data 8).

We further show that the inbreeding coefficient was more strongly correlated with $H_0$ than with heterozygosity near Copia and Harbinger TEs ($H_{TE}$, Fig. 3c and Supplementary Data 6), suggesting that Copia and Harbinger TEs cause deviations from the expected

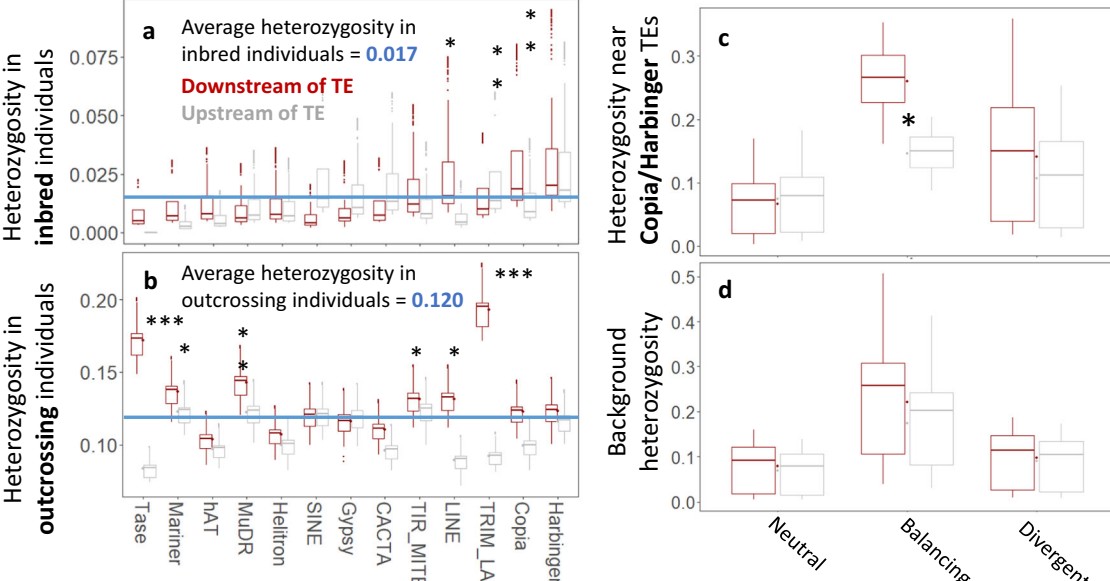

**Fig. 3 | Heterozygosity across TE superfamilies in inbred and outcrossed individuals.** Heterozygosity in inbred individuals (genome-wide $F_{IS} > 0.6$) was generally low but was elevated near Copia, Harbinger and LINE elements (**a**). While Copia and Harbinger elements can maintain heterozygosity particularly under inbreeding, only LINE elements are also associated with increased heterozygosity in outcrossing populations (genome-wide $F_{IS} < 0.6$) (**b**). We therefore considered the Copia and Harbinger elements to contribute to the "TE Effect" (Table 1). As the CACTA superfamily was associated with intermediate levels of heterozygosity in the American *A. lyrata* lineage, we used CACTA as the reference group (intercept) for statistical comparisons with other TE superfamilies in the mixed model output of Supplementary Data 6. Heterozygosity also depended upon the signature of selection (**c**, **d**), with balancing selection typically increasing heterozygosity, particularly downstream of TEs. Background heterozygosity ($H_0$) refers to heterozygosity near all TE superfamilies but Copia and Harbinger. Significance of factor level estimates (*, ** represent p-values < 0.05 and 0.01 obtained by linear mixed models without correction for multiple testing, resp.) are provided in Supplementary Data 6. Panels **a**–**c** and are based on 6936, 10404, 1820, and 10318 independent data points, respectively (i.e. SNPs across genetically distinct clusters). Centre, bounds, and whiskers of boxplots represent median, Q1/Q3 quartiles and lower/upper bounds, respectively. Source Data are provided as a Source Data file.

relationship between heterozygosity and $F_{IS}$. Elevated levels of balancing selection downstream of these two TE superfamilies (Fig. 4a, b, $F = 1.83^*$, Supplementary Data 6) appear to be responsible for the relatively high heterozygosity levels downstream of Copia and Harbinger TEs in populations subject to intense inbreeding (Figs. 3a and 4c). Together, these findings (Figs. 3 and 4) support our hypothesis that TEs can provide evolutionary potential under inbreeding (Fig. 1-Q2).

In line with our final hypothesis that the role of TEs in evolution is consistent across sampling areas (Fig. 1-Q3), we finally found that TE-specific signatures of selection (Fig. 4c) were reproducible across evolutionary lineages. First, proportions of outliers near TEs per superfamily correlated between North-American and European populations (Fig. 5a and Supplementary Data 9). However, this correlation was positive for outliers upstream of TEs (Pearson's $r$ 0.54* and 0.26 for balancing and divergent outliers, resp.) and negative for outliers downstream of TEs (Pearson's $r$ −0.49(*) and −0.57* for balancing and divergent outliers, resp.). It is quite remarkable that TE-related signatures of selection were correlated between sampling areas with distinct evolutionary histories and environmental conditions, the latter being reflected by the low number of SNPs sharing signatures of selection between the two sampling areas (Fig. 5b). Second, heterozygosity near Copia and Harbinger elements ($H_{TE}$) did not significantly differ from background heterozygosity ($H_0$) in outbreeding European samples with $F_{IS} < -0.3$ (Fig. 5c), which is in accordance with the hypothesis that the association of Copia and Harbinger elements with heterozygosity is inbreeding-dependent. The TE effect was, however, marginally present in outcrossing European populations with $F_{IS} > -0.3$ (Supplementary Fig. 3 and Supplementary Data 9), suggesting that demographic processes rather than mating system strategy drive the TE effect on heterozygosity under intense inbreeding.

## Discussion

Despite more than a century of research on the molecular basis and ecological consequences of inbreeding, we still poorly understand why many natural populations characterized by intense inbreeding can frequently continue to evolve. While mutations are the predominant driver of evolution in the absence of outcrossing, they typically arise at low rates that are unlikely to maintain evolution[43]. Here, we capitalize on the known mutagenic properties of TEs to assess their role in evolution across an inbreeding gradient in the mixed mating species *A. lyrata*. We found that (i) heterozygosity is elevated downstream of TEs, (ii) this increase in heterozygosity is associated with housekeeping functions, and may thus provide a basis for evolution under inbreeding, and (iii) the role of TEs in adaptive evolution is consistent across independent evolutionary lineages. In general, TEs appear to be involved in heterozygote advantage that becomes particularly apparent under inbreeding, and in frequency-dependent balancing selection through their overrepresentation in gene ontology processes related to signaling and (a) biotic stress responses. Our findings provide important indirect evidence for TEs being responsible for much of the evolutionary trajectories of natural populations with reduced outcrossing rates, and that genomic regions downstream of TEs provide functional genetic variation as potential targets of adaptive evolution.

As methylation and mutation rates have been found to be elevated up to 3.5 kb away from a TE[37,62,63], most of the SNPs identified here (65%) appear to be within reach of TEs (Figs. 2a, c). Interestingly, however, while mutation rates have been shown to decrease gradually with genomic distance from TEs[37], we don't observe equivalent effects on heterozygosity. We suspect this finding to result from TE insertion preference and host-TE co-evolution driving the evolutionary fate of mutations. Specifically, TEs tend to insert (i) near functional genes, where mutations flanking TEs in particular can provide fitness benefits to the host[41,55], and (ii) away from vital genes, where mutations more

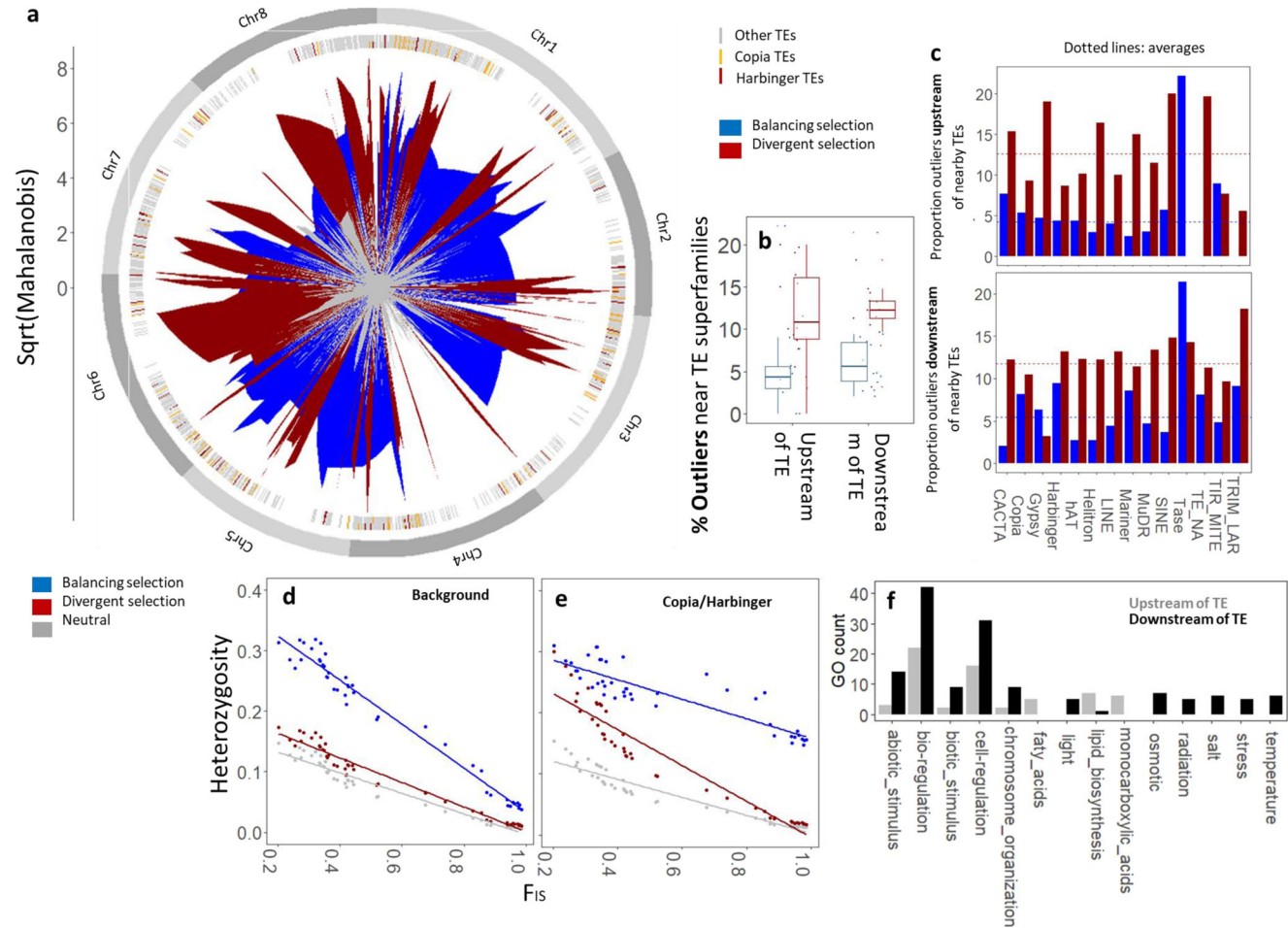

**Fig. 4 | Genome-wide signatures of selection across TE superfamilies.** The Mahalanobis distance represents the deviation from neutrality (**a**), with 4.86% and 12.17% of genome-wide SNPs manifesting signatures of balancing and divergent selection, respectively (Supplementary Data 1). Genome-wide signatures of divergent and balancing selection are similar upstream and downstream of TEs (**b**), but vary considerably among TE superfamilies (**c**). As compared to background heterozygosity ($H_0$), Copia and Harbinger elements are associated with elevated levels of nearby heterozygosity under inbreeding ($H_{TE}$) for SNPs under balancing selection (**d**, **e**). While signatures of selection are similar upstream and downstream of TEs, SNPs downstream of TEs are consistently implicated in biological processes related to stress (**f**). Significance of factor level estimates are provided in Supplementary Data 6. Balancing selection is based on negative alpha values in Bayescan analysis (loci with significantly reduced genetic differentiation), and is distinguished from signatures of purifying selection because they generally represent high-frequency variants (see Supplementary Fig. 2). Panels **a**–**d** are based on 1734 independent SNPs. Panels **d** and **e** are based on 10318 and 1820 independent data points, respectively (i.e. SNPs across genetically distinct clusters). Panel f is based on 203 annotated SNPs. Centre, bounds and whiskers of boxplots represent median, Q1/Q3 quartiles and lower/upper bounds, respectively. Source Data are provided as a Source Data file.

distant from TEs but residing within these vital genes can be harmful or beneficial to organismal fitness depending on their impact on gene functioning. As a result, the genome-wide distribution of proximity effect of TEs on heterozygosity likely varies with gene function and essentiality. The tendency of SNPs with signatures of balancing selection downstream of TEs to appear in essential genes (Supplementary Fig. 2) contributes to the notion that TE-host co-evolution involves genomic regions essential to organismal functioning. Importantly, while mutations arising in essential genes that are vital to organismal functioning are typically considered harmful and the target of purifying selection, our results suggest that mutations occasionally persist in essential genes where they can contribute to balancing evolution.

Extreme inbreeding levels typically increase the risk for inbreeding depression, i.e. the loss of functioning due to recessive deleterious mutations becoming homozygous to such an extent that fitness is compromised[10,12,13]. Our findings suggest that TEs can counteract this loss of functioning, through generating genetic variation in fitness-related genomic regions containing vital genes directly impacting fitness or organismal functioning. Specifically, heterozygosity remained high under intense inbreeding near Copia and Harbinger elements

(Figs. 3a and 4e), and these heterozygous genetic variants were enriched for molecular processes that are vital to organismal functioning (e.g. RNA processing, protein transport and response to hormones; Supplementary Data 8). Although experimental validation is required to confirm this heterozygosity enhancer activity observed in our study, the heterozygote advantage expected for fitness-related genomic regions that may be sensitive to inbreeding depression is corroborated by the significant role of balancing selection in maintaining heterozygosity under inbreeding (Fig. 4e). Interestingly, our findings may help explain earlier studies showing that self-fertilizing *A. lyrata* populations do not exhibit reduced fitness as compared to outcrossing populations[64,65]. The potential existence of TE-related molecular strategies maintaining heterozygosity at fitness-related genomic regions represents a compelling line of research that can explain the lack of inbreeding depression under intense inbreeding.

Transposable elements have frequently been associated with fitness and adaptive evolution[39,66–68]. However, while TEs can be directly targeted by selection[67–70], they may also facilitate selection across the genome through increasing mutation rates in functional and stress-responsive genomic regions[33,35]. We observed genome-wide

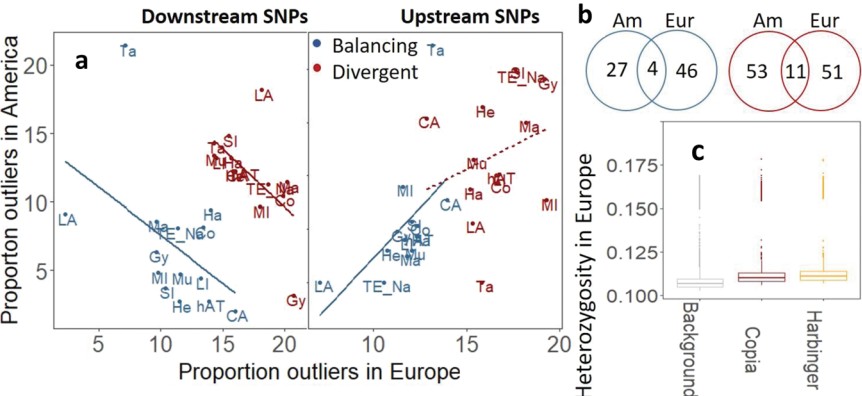

**Fig. 5 | Signatures of selection across *A. lyrata* lineages.** The proportion of outliers associated with TE superfamilies correlates between the two sampling areas (**a**), even though few SNPs shared signatures of selection between the two sampling areas (**b**). $H_0$ (background heterozygosity) is not significantly different from $H_{TE}$ (heterozygosity associated with Copia and Harbinger elements) across outcrossing European *A. lyrata* genomes (**c**, Supplementary Data 9), confirming that the effect of Copia and Harbinger elements on nearby heterozygosity is limited to inbred populations. Blue and red colors represent signatures of balancing and divergent selection, respectively (**a**, **b**). Balancing selection is based on negative alpha values in Bayescan analysis (loci with significantly reduced genetic differentiation), and is distinguished from signatures of purifying selection because they generally represent high-frequency variants (see Supplementary Fig. 2). Panel **c** is based on 12,138 independent data points (1734 SNPs across seven genetically distinct clusters). Source Data are provided as a Source Data file.

heterozygosity enhancer activity for genetic variants downstream of their nearest TE, where SNPs are enriched for gene ontology processes involved in stress responses and regulation of biological traits (Supplementary Data 8). The association of SNPs downstream of TEs with stress responses likely points to the role of many co-opted TEs as regulators of nearby stress responsiveness[49,51,52,71]. While increased levels of selection were not observed downstream of TEs, the maintenance of genetic variation in stress-responsive genomic regions may result from fluctuating stress levels caused by a variety of environmental factors, and represents a basis for adaptive evolution when environmental conditions change. Schrader et al. correspondingly demonstrated increased evolutionary rates in genomic regions featured by high densities of TEs, most likely contributing to rapid adaptation in genetically depleted, invasive ant populations[33].

Here, we add to this rare evidence of genome-wide impacts of TEs on evolution through (i) demonstrating that TEs are implicated in adaptive evolution both in inbreeding and outcrossing plant populations, and (ii) revealing a potential association between specific TE superfamilies and the sensitivity of populations to inbreeding depression. Specifically, we found TE superfamily-specific signatures of selection that could be replicated across independent sampling sites (Fig. 5a). Although more outliers were found in Europe (Fig. 5a, b), which is likely due to the more extensive geographic sampling scale, superfamilies with limited upstream signatures of selection across North-American populations had correspondingly limited signatures of upstream selection across European populations and vice versa (Fig. 5a). Evolutionary trajectories of populations thus seem to be associated with TE superfamilies integrating at non-random positions in the genome.

Genetic variants downstream of TEs show a notably distinct pattern as compared to upstream signatures of selection. Specifically, where TE superfamilies are featured by a high proportion of downstream outliers in North America, they typically have a low proportion of downstream outliers in Europe, and vice versa (Fig. 5a). Interestingly, a GO enrichment analysis of downstream genetic variation for each of the TE superfamilies with at least 15 downstream SNPs (Supplementary Data 8) revealed that each TE superfamily is enriched for distinct molecular and biological processes. We found that genetic variation downstream of (i) MuDR TEs was involved in photosynthesis and growth, (ii) Helitron was associated with abiotic stresses (heat, light, drought), and (iii) MITEs with wounding and disease (Supplementary Data 8). This compartmentalization of selective signatures near distinct TE superfamilies may explain the negative correlation in selection signatures downstream of TEs for American vs. European sampling areas, which are featured by distinct evolutionary histories and selective pressures.

While calling for further research, our results collectively suggest that each TE superfamily can facilitate evolution in particular environmental and demographic situations. Specifically, where Copia and Harbinger elements are associated with increased heterozygosity in housekeeping genes, presumably to maintain fitness under elevated levels of inbreeding, other TE superfamilies are linked to genetic diversity in functional genomic regions and may thus promote adaptive evolution under changing environmental conditions. Climate warming may, for example, drive adaptive evolution particularly downstream of Helitron elements, which were found to be associated with genetic variants involved in heat and drought responses. Helitron elements have correspondingly been shown to manifest differential expression in *A. arenosa* across independent elevation gradients[71], corroborating their involvement in climate adaptation. We suggest that the combined actions of TE mutability, purging and natural selection may represent a powerful evolutionary strategy allowing long-term persistence of naturally inbred populations. More broadly, this theory could explain the success of self-fertilizing populations across the globe. Importantly, this does not refute the idea that self-fertilizing populations frequently are evolutionary dead-ends arising from continuous drift-induced accumulations of deleterious mutations[72,73], but see ref. 74. Specifically, while rates of extinction have been found to be significantly higher in self-pollinating as compared to outbreeding taxa[75,76], self-pollinating taxa that do persist through time may do so through evolving strategies counteracting inbreeding depression.

Our findings do not suggest a significant role for purifying selection and mutational meltdowns as drivers of genetic variation across the genome of self-fertilizing *A. lyrata*. While lethal TEs are rapidly purged from a population[41], slightly or conditionally deleterious alleles arising near TEs are more difficult to purge, and may reach higher frequencies resembling signatures of balancing selection[77]. The lack of low-frequency signatures of balancing selection (Supplementary Fig. 2) suggests that purifying selection does not considerably contribute to genome-wide variation across American *A. lyrata* populations, with the enrichment of processes related to signaling and biotic responses particularly supporting a role for frequency-dependent balancing selection (Supplementary Data 8). In *A. thaliana*,

genome-wide signatures of balancing selection are often associated with resistance and stress responses[78], and with temporally fluctuating selection pressures acting upon defense metabolism and signaling genes[79]. Note that pseudo-overdominance (at deleterious loci[80]) and associative overdominance (at neutral loci[81]), both arising from purifying selection at linked recessive deleterious loci, may also contribute to genome-wide signatures of balancing selection. Both types of selection are covered by our analysis of purifying selection (see Supplementary Fig. 2), with pseudo-overdominance and associative overdominance generating signatures of purifying selection (negative alpha values, low maf, predominant non-synonymous codon usage) and apparent purifying selection (negative alpha values, low maf, variable codon usage), respectively. Although fully disentangling the various types of overdominance remains challenging and some of our 'real' signatures of balancing selection may actually reflect pseudo-overdominance[80], our findings suggest that pseudo-overdominance and associative overdominance are strongly underrepresented in our dataset, particularly for the American lineage (Supplementary Fig. 2).

The patterns observed in this study likely underrepresent the actual role of TEs in driving genome-wide variation and evolution. First, it is unlikely that the annotated TEs are present across the sampling range, and their influence in driving adaptive evolution at nearby genes is limited to the samples actually harboring the TEs. It can nevertheless be expected that TEs conferring important fitness benefits are more likely to spread across the host range. We correspondingly suspect that Copia and Harbinger TEs in particular may be ubiquitous across the range due to their presumable role in housekeeping gene functioning. Recent findings in *A. thaliana* confirm that non-reference TE insertions appear at low frequency due to their deleterious nature, unless they integrate in functional genomic regions where natural selection facilitates the spread of TEs[41]. Second, many TEs cluster into TE islands, corresponding to the idea that functional genomic regions are frequently revisited by TEs[41]. In TE islands, the associations between heterozygosity at specific genetic variants and their nearest TE may be impacted by another nearby TE. Such TE interactions may have reduced our ability to detect significant effects of individual TEs. Further research focusing on the relationship between the abundance of specific TEs across the range and their potential to facilitate heterozygosity and evolutionary dynamics at nearby genes may corroborate the underappreciated role of TEs as enhancers of functional and even vital genetic variation across the genome.

## Methods

### Sampling, SNP calling, and TE annotation
The sampling of *A. lyrata* populations from the North American Great Lakes region and Northern Europe is fully described in Buckley et al.[82]. Briefly, seeds from individual plants (representing distinct maternal families) were sampled from field sites and grown in standard greenhouse conditions to produce leaf tissue for DNA extraction[82]. Only confirmed diploid populations are used[83]. Where seeds were not available, dried leaf tissue sampled from the field was used[82]. RAD-seq genotyping using Illumina sequencing (via Edinburgh Genomics) was then performed to generated single nucleotide markers (SNPs) for 91 individuals from 13 North American populations ($N = 4$ for each of 12 populations, except one population where $N = 1$, total $N = 49$) and 2–3 individuals from each of 18 Northern European populations (total $N = 42$; see Supplementary Data 5 for details)[84]. The level of inbreeding and therefore genome-wide heterozygosity varies significantly across North American populations, which was in turn lower than heterozygosity estimates across Northern European samples[84].

The program Stacks was used to call SNPs across 92 bp RAD loci aligned to the *A. lyrata* reference genome. Filtering was applied to remove incorrectly assembled loci and reduce the likelihood of incorporating genotyping errors (using only loci with a minimum read depth of 10). For analyses, we randomly selected only one SNP per RAD locus and only SNPs present in all individuals (so unlinked SNPs without missing data). We limited the dataset to highly qualitative SNPs through maintaining (i) minimum allele frequencies of 0.05, (ii) no missing data, and (iii) an average read depth of 73.18 (European samples) and 60.09 (North American samples) per SNP (Supplementary Fig. 4). This resulted in a dataset of 5935 RAD loci (and thereby 5935 SNPS) across the two *A. lyrata* subspecies. Extracting those present in just North American or European populations, resulted in 1858 and 4398 SNPs respectively (Supplementary Data 1 and 2).

TEs annotations were obtained from Legrand et al.[54]. Briefly, libraries of consensus sequences representing repetitive elements were obtained from genome assemblies of *A. thaliana*, A. *lyrata*, and *A. halleri* using the TEdenovo pipeline of the package REPET[85]. A bundle library was then obtained by pooling these libraries and was used to annotate TEs in the *A. lyrata* assembly also used for RAD sequencing (Phytozome v11), using the TEannot pipeline of REPET. We calculated the genomic distance of each SNP to its nearest TE and the orientation of the SNP relative to its nearest TE (within TE vs. downstream vs. upstream). Through focusing on nearest neighbors, the probability of non-reference TE insertions (which typically arise at low frequency and randomly across samples[41] even closer to our SNPs is strongly reduced.

### Inbreeding coefficients and signatures of selection
The genome-wide inbreeding coefficient ($F_{IS}$) was calculated per sample using the R package "hierfstat" (see Supplementary Data 3 and 4 for input files). We further applied two approaches for detecting signatures of adaptive evolution associated with each TE superfamily. First, a multivariate approach implemented in PCadapt was used for the dual purpose of estimating relatedness among samples (multi-dimensional genetic background structure) and detecting signatures of natural selection identified as significant deviations from background genetic structure[86,87]. Based on the distribution of eigenvalues associated with the principal components (PCs) (Supplementary Data 7), we selected the first three PCs to represent background genetic structure and among-sample relatedness. The strength and significance of selection were expressed as Mahalanobis distance (multi-dimensional distance of SNPs from genetic background structure), with $q$-values below 0.01 representing significant signatures of selection. Second, Bayescan was used to identify signatures of balancing selection, with Log10 values of the posterior odds (PO) > 0.5 representing 'substantial' evidence for selection[88]. Bayescan decomposes $F_{ST}$ coefficients into population-specific (beta) and locus-specific (alpha) effects, with negative alphas pointing to balancing or purifying selection. The false discovery rate (FDR) was set at 0.05. For each TE superfamily, we calculated the proportion of SNPs with signatures of balancing (low $F_{ST}$) vs. divergent selection (high $F_{ST}$).

To assess the role of purifying selection in producing negative alpha values, we explored the tendency of SNPs with negative alpha values to be associated with low-frequency non-synonymous variants. If alleles at these loci were rarer than expected under neutrality, we considered these SNPs candidates of purifying selection[77,89,90]. Specifically, we considered SNPs with signatures of balancing selection (negative alpha values in Bayescan analysis) as candidates of purifying selection if their minimum allele frequency (MAF) was (i) smaller than the MAF of neutral SNPs, and (ii) smaller for non-synonymous as compared to synonymous codon usage. In any other cases, we considered SNPs with negative alpha values as candidates of balancing selection. Codon usage (4-fold and 0-fold positions in the *A. lyrata* reference genome) of individual SNPs was obtained using the NewAnnotateRef.py script[91].

### Modeling the effects of TEs on heterozygosity and evolution
We followed a stepwise modeling approach that allowed testing our hypotheses consecutively without running complex models with many

**Table 1 | List of generalized mixed models used to test our three main hypotheses**

| Model | | Hypothesis (Fig. 1) |
|---|---|---|
| 0a ($F_{IS} > 0.6$) | H ~ TE Superfamily:Sense + Distance | Explorative |
| 0b ($F_{IS} < 0.6$) | H ~ TE Superfamily:Sense + Distance | Explorative |
| 1 | H ~ Distance×TE Effect + TE Effect×Sense + Distance × Sense | Q1 |
| 2a | $H_{TE}$ ~ $F_{IS}$ × Selection + Sense × Selection | Q2 |
| 2b | $H_O$ ~ $F_{IS}$ × Selection + Sense × Selection | Q2 |
| 3 | H (Europe) ~ TE Effect + Sense | Q3 |

The probability of a locus to be heterozygous was consistently modeled with the random effect "Relatedness", a categorical variable correcting for genetic relatedness among samples as revealed by a multivariate PCadapt analysis. TE superfamilies with significant vs. no impact on heterozygosity under inbreeding (Model 0) were grouped into the binary grouping variable "TE Effect". We tested both a linear and quadratic effect of the "Distance" between a SNP and its nearest TE on its heterozygosity. See Fig. 1 for an overview of the three corresponding hypotheses.

parameters. We first ran an explorative mixed model (Model 0, Table 1) to: (i) identify TE superfamilies with elevated levels of heterozygosity under inbreeding, and (ii) reduce data complexity in subsequent models through replacing the TE superfamily variable ($N = 13$ levels) by a simple TE grouping variable ("TE Effect", $N = 2$ levels). TE effect thus refers to the potential of specific TE superfamilies (here Copia and Harbinger elements) to maintain nearby heterozygosity under inbreeding. This new grouping variable distinguishes between TE superfamilies that significantly impact heterozygosity under inbreeding ($H_{TE}$) and all other TEs ($H_O$). To test whether this effect was exclusively associated with high inbreeding levels, the explorative model was run on a subset of the samples characterized by predominant self-fertilization and with inbreeding coefficients exceeding 0.6 (Model 0a), and compared to a model with subset of the samples characterized by predominant outcrossing and with inbreeding coefficients below 0.6 (Model 0b). The probability of a locus to be heterozygous was consistently modeled with the random effect "Relatedness", a categorical variable correcting for genetic relatedness among samples as revealed by the multivariate PCadapt analysis. Throughout our study, the probability of a locus to be heterozygous was modeled using a binomial distribution with cloglog link within the *glmmTMB* R package[92]. Analysis of variance tables were produced using the *lmer* R package.

Model 1 (shown in Table 1) was built to test our first hypothesis (Q1 in Fig. 1), i.e. how does the position of a SNP relative to its nearest TE affect its heterozygosity? To this end, we explored the interactive effect between "Distance" and the orientation of a SNP relative to its nearest TE ("SNP_sense": downstream vs. upstream of TE vs. within TEs) on the sensitivity of a SNP to changes in heterozygosity. "Distance" was included to explore the tendency of SNPs to be heterozygous with decreasing distance to the nearest TE (linear) or at intermediate distance (quadratic). We hypothesized the relative position of a SNP to have a larger effect on $H_{TE}$ than on $H_O$, and therefore also included "TE Effect" in interaction with "Distance" and "SNP_sense" in Model 1 (Table 1).

Model 2a (Table 1) relates to our second hypothesis (Q2 in Fig. 1). Specifically, we tested whether nearby heterozygosity ($H_{TE}$) caused deviations from the expected negative relationship between heterozygosity and inbreeding, and whether this deviation was associated with signatures of selection. The variable "Selection" indicated whether a SNP was an outlier of balancing selection or of divergent selection rather than a neutral SNP (three levels). To test whether the patterns observed for Model 2a are unique to the TE superfamilies identified in Model 1, we compared the Model 2a outcomes to a competing background model (Model 2b) ran on all other TE superfamilies. We expected the inbreeding coefficient to show a stronger

correlation with $H_O$ (Model 2b) than with $T_{HE}$ (Model 2a, cfr. Q2 in Fig. 1). We preferred competing models over one all-inclusive model to avoid three-way interactions (fictional Model 2: H ~ $F_{IS}$ × Selection × TE Effect + Sense × Selection × TE Effect). We expected that (i) increased heterozygosity near TEs would be associated with signatures of balancing selection facilitating heterozygosity under inbreeding, particularly downstream of TEs and for $H_{TE}$, and (ii) some TE superfamilies may facilitate divergent selection driving SNPs towards fixation (i.e. reduced heterozygosity) but this effect should be more pronounced for $H_O$ than for $H_{TE}$.

Model 3 (Table 1) was built to test our third hypothesis (Q3 in Fig. 1) assessing the TE effect (impact of TE superfamilies on heterozygosity under inbreeding) on an evolutionary and spatially independent *A. lyrata* lineage that is characterized by an outcrossing mating system. If the role of particular TEs in evolution is particularly important under inbreeding, the reinforcing effects of these TEs on heterozygosity across inbred populations should be absent across outbred populations. In addition, if TEs are involved in evolution irrespective of inbreeding levels, signatures of selection associated with TEs identified across the North-American samples should be predictive for evolutionary signatures in other populations (Q3 in Fig. 1). We correspondingly formulated two specific expectations on heterozygosity for an independent European dataset where inbreeding is strongly reduced, with inbreeding coefficients ranging from −0.24 to −0.64[82,84].

First, $H_O$ and $H_{TE}$ should be comparably high for the European dataset (as opposed to the inbred samples originating from North-America where $H_{TE}$ by definition is higher than $H_O$). We thus modeled the "TE effect" on heterozygosity across European *A. lyrata* populations, while accounting for potential effects of SNP orientation relative to nearest TE. We expect no TE effect on $H_{Europe}$ (i.e. $H_O = H_{TE}$). To disentangle mating strategy effects from demographic effects on the inbreeding coefficient, we explored this expectation on two subsets that share the same outcrossing mating system but differ in inbreeding coefficients ($F_{IS} > −0.3$ vs. $F_{IS} < −0.3$), presumably due to strong demographic bottlenecks experienced by the most inbred European individuals[93]. If mating system strategy is the driver of TE effects, than $H_O = H_{TE}$ for both subsets. If demography drives TE effects, than $H_O = H_{TE}$ only for the subset with $F_{IS} < −0.3$. Second, the proportion of SNPs under selection across TE superfamilies identified in the North-American dataset should be comparative for the European dataset. While very different outliers of selection are expected in Europe (distinct evolutionary histories and strong environmental differences), the general tendency of TE superfamilies to be involved in specific evolutionary processes may be similar between the North-American and the European samples. To test this, we compared the proportion of outliers of each TE superfamily between the North-American and the European dataset (Pearson correlation). We applied the same outlier detection approaches used for the North-American dataset to the European accessions.

All analyses were performed at the superfamily level because (i) the number of replicate TEs within families was too low to statistically analyze heterozygosity at the TE family level, and (ii) it allows to elucidate genome- and superfamily-wide relationships between TEs and nearby genetic diversity.

## Gene ontology and essentiality analysis
We used Blast2go to map and annotate the RAD loci containing the SNPs against all plant genomes to obtain gene ontology terms. With Fisher's exact tests (R package "TopGO"), we then assessed which biological processes and molecular functions were particularly overrepresented in (i) outliers of selection, (ii) SNPs near TE superfamilies that significantly increase heterozygosity under intense inbreeding ($H_{TE}$), and (iii) SNPs upstream vs. downstream of TEs. We hypothesized that outliers of divergent selection, and SNPs downstream TEs are enriched for stress-related processes (e.g. response to abiotic

stimulus, regulation of phenotypic traits). Outliers of balancing selection and SNPs underpinning $H_{TE}$ are hypothesized to be enriched for fitness-related and housekeeping processes (e.g. disease, protein folding, nucleotide metabolism, and intracellular transport).

For all SNPs within genes, we calculated two proxies of gene essentiality and tested whether SNPs near TE superfamilies are particularly essential to organismal functioning, depending on their signature of selection. First, we assessed the size of the gene family, with single copy genes tending to be more essential due to the lack functional redundancy[54]. Second, we investigated the ratio of non-synonymous to synonymous substitution rates (Ka/Ks), which is expected to be lower for more essential genes[54]. An analysis of variance (ANOVA) was used to compare these measure of gene essentiality between signatures of selection (balancing, divergent, and neutral) and depending on orientation relative to TE (downstream, upstream).

## Reporting summary
Further information on research design is available in the Nature Portfolio Reporting Summary linked to this article.

## Data availability
Previously published transposon dataset under BioProject PRJNA495003 were used in this study. In addition, the previously published raw demultiplexed FASTQ files can be found in NCBI SRA database under accession SRP148549 and the IDs for the demultiplexed FASTQ files for individual BioSamples are SAMN09230090 - SAMN09230180. Source data are provided with this paper.

## Code availability
Customized R code is available at Github [https://github.com/hannedekort/lyrata].

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

## Acknowledgements

This work was supported by Research Foundation Flanders (grant numbers FWO 12P6517N, FWO 12P6521N), and by the Natural Environment Research Council (NERC project grant NE/H021183/1 to Barbara Mable, which generated the RAD data). We thank Edinburgh genomics for conducting the RAD sequencing. We also thank Dr. Pierre Baduel for his expert insights on the evolutionary importance of reference vs. non-reference TEs in *A. thaliana*.

## Author contributions

HDK formulated the hypotheses, performed most analyses, and wrote the manuscript. SL provided the TE data and performed codon usage and essentiality analyses. OH fine-tuned the context and conservation implications of the study. JB provided the filtered SNP data and all information on the *A. lyrata* samples. All authors provided feedback to the first versions of the manuscript.

## Competing interests

The authors declare no competing interests.
