## [Peer Review File · Nature Communications]

Transposable elements maintain genome-wide heterozygosity in inbred populationsReviewers' Comments:

Reviewer #1:

Remarks to the Author:

This paper examines whether transposable elements play a role in "generating" genetic variation (heterozygosity) through increased mutation rates, thereby increasing adaptive potential. Specifically, the authors test whether highly inbred populations may benefit from this process (as a mechanism to compensate for the reduced overall genetic diversity due to inbreeding).

Using a RAD-seq approach, authors find evidence for elevated heterozygosity downstream of some TE superfamilies, and indications that this increased heterozygosity may be associated with functional genes through detection of signatures of selection (outlier detection) and gene ontology/enrichment analyses. In general, I think the paper addresses a timely and original question, takes an original approach and has a message that is of broad interest.

However, it was not an easy read, and perhaps because of this, I have several major concerns, mostly related to the methodology.

Overall, I think the genomic analyses (i.e., the genotyping and TE identifications) were thorough. The only aspect I wonder about is whether the minimum read depth of 10 per called locus may have been on the low side? Authors may want to justify this choice in the paper. Related to this, please specify whether there was a minimum allele frequency required for a SNP to be called? Looking at the supplemental tables, SNPs are scored as 0 (hom) or 1 (het). Surely, for Bayescan, the actual allelic states were used? It would be good to provide these in a data repository, if this has not been done so already (in which case, please mention the link).

The statistical analyses were more difficult to follow, and I identified a few potentially major concerns:

- 1) Writing and presentation. Both the results and methods sections were hard to follow. In the results section, subheadings would be helpful to give some structure. Also, I would avoid too many almost literal stylistic repetitions of intro and methods, but rather formulate the results as findings, with a brief interpretation in relation to the hypotheses. For example: In support of *idea/hypothesis*, we found XX was related to YY.

- 2) Bayescan – have authors ignored purifying selection? Especially in the results, but also the methods, I think the Bayescan approach should be described better. The results strongly rely on the Bayescan outlier detections. However, the resulting classes of loci (under balancing selection, under divergent selection & neutral) are very much taken as a given for making certain comparisons. It took me a while to figure out that these classes had been inferred with Bayescan (probably based on the alpha parameter, but this should be explained in the paper, neither the methods L351-354 mention Bayescan, nor do the figure captions). A major concern related to this is the interpretation of outlier loci with $\alpha < 0$ as under balancing selection. In my understanding (which is limited, so please rebut if I missed something obvious) it is not really possible with Bayescan to distinguish between signatures of balancing vs. purifying selection. Thus, unless the authors have taken an analytical step enabling them to rule out purifying selection (if so, I don't think they have described it in their paper), their classification of loci with $\alpha < 0$ as being under balancing selection is misleading. I would personally expect strong signatures of purifying selection, as one would expect most mutations (including those due to TE activity) to be deleterious. If I am correct, this would somewhat weaken the overall author's interpretation that TE-associated mutations increase evolutionary potential, in which case the paper's general message should be toned down

- 3) Generalized linear mixed modelling. There are a couple of very confusing aspects to the modelling approached described:

- a) Relatedness seems to be a continuous variable but has been included in the model as a random effect. However, AFAIK, random effects are always categorical. So, it seems to me that relatedness

should have been included as covariate. Despite the explanations in L318-320, I could not understand how it was calculated. What is a "sample" in this context? Relatedness to what? Is it simply the groupings arising from the PCA?

b) I do not understand the logic of the competing background models, and I think it has not been explained how this works at all. Why is it better or more appropriate than using a single model with a binomial error term to explain the probability of heterozygosity (H) including fixed terms: TEsuperfam, FIS, Selection, Poly(TEdist quadratic), TEsense, SNPsense and (a choice of) their interactions?

I am sure there is good reasons for the taken approach, but the reader needs much more guidance. It is currently impossible to judge whether the taken modelling approach is appropriate or not.

4) FIS. The authors use FIS as an estimation of inbreeding levels, and it is reassuring that the highest values correspond to populations known to display high selfing rates. Still, I disagree that we truly have a gradient here. In line with what we know from earlier work, most populations are either highly outcrossing, or highly selfing. The FIS values reflect this, as data points are really sparse between $FIS=0.5$ and $FIS=0.9$. The patterns in Fig 4 D and E remain extremely interesting regardless, with clearly increased heterozygosity at all FIS levels for loci under purifying (or balancing) selection.

5) Tests of significance. Authors mainly report F values, but they seemingly mention only one degree of freedom (I assume the residual df are not mentioned). For example - L136 $F_{13}=4.98^{**}$. Meaningless if we do not know the residual df. Moreover, the * are not explained (I assume ** means $p<0.01$). In my understanding, F values indicate whether fixed effects are significant or not. However, if such fixed effects have multiple levels, they do not inform about differences between the factor-levels, for which post hoc comparisons are required. It only became clear after looking at Table S5 that in reality the significance of factor level estimates are evaluated (z-tests).

6) Related to the previous point, I struggled to understand what the intercept represented in the models. For example, there are 13 TE superfamilies in figure 3A, and 13 factor levels listed in Table S5, plus an intercept (i.e., 14 levels). It is probably my ignorance, and has a very simple explanation, but it would be good to include this explanation in the paper.

Likewise, I struggled to understand how a binomial/binary Y can be zero-inflated.

7) TE superfamily. Could an individual SNP be affected by more than one TE superfamily? I think the models assume that SNP heterozygosity can only be affected by its closest TE, but how realistic is this? Could there be interactive effects of TEs?

I also have a few conceptual concerns:

1) Inbreeding depression/purging as a strategy.

In my view, purging is not a strategy, it just happens, as a consequence of inbreeding. L45 – deleterious alleles may be purged, but not complete loci. In the discussion, there is a misconception that repeated selfing increases the risk of inbreeding depression (L214-216). Given purging, the opposite is true.

2) The motivation for this study is understanding "the evolutionary success of selfers". While I completely agree that it is intriguing that selfers manage to occupy a wide range of habitats, they might not necessarily achieve this through adaptation. The alternative view – that selfers are evolutionary dead-ends, but may evolve repeatedly due to short term advantages – also requires attention. Under this scenario, selfers do not need to adapt per se. They may arise pre-adapted to the prevailing conditions (from a locally adapted outcrossing ancestor), or many formed selfing lineages may arise, after which selective filtering allows establishment of those lineages that are best-adapted to the prevailing conditions. Either way, such lineages may eventually go extinct when those conditions change due to limited adaptive potential. It would be good to discuss whether the observed patterns of increased heterozygosity in highly inbred individuals could actually reflect mutational

meltdown rather than increased adaptive potential. This would at least explain the finding of many loci being under purifying selection (interpreted by the authors as balancing selection). The work by Emma Goldberg et al (Science 2010) comes to mind. In any case, I would play down L191-194 a bit.

Minor remarks

Fig 1 – Are any of the European samples polyploid? If so, does this affect the outcome in any way?

L229 – balancing SNPs?? Do you mean SNPs with signatures of balancing selection (or rather purifying selection)?

L299 – please indicate where you obtained this reference genome, as there are multiple versions circulating.

L302 – “only those” – only those SNPs or only those RAD loci?

L232-239 This paragraph feels like a loose end not tied in with any findings. Should it be integrated into the next paragraph?

Reviewer #2:

Remarks to the Author:

In this manuscript by De Kort and colleagues the authors have reanalyzed RADseq data from 91 individuals of *Arabidopsis lyrata* (from US and EU, a total of around 5900 SNPs) with contrasted breeding systems, in order to test the potential role of transposable elements (TEs) in maintaining genetic variation in inbreeding conditions. While the biological hypothesis is very novel and promising, I see maybe one major drawback in this study. The study investigates the proximity of SNPs with given TEs based on the *lyrata* genome annotated with REPET. However it is well documented that a plethora of TE insertion polymorphisms (TIPS) are found in plant populations, even in narrow geographical areas (see in Nature Comm 2019 the works by Quadrana et al for *Arabidopsis thaliana* <https://doi.org/10.1038/s41467-019-11385-5> and Carpentier et al, for *Oryza sativa* <https://doi.org/10.1038/s41467-018-07974-5> for instance). In this context, how can the authors estimate the precise location of the TE families they investigate?

Minor comments:

Line 209: Overstatement: The simultaneous enrichment of biological regulation and stress-responsiveness found here suggests that TEs are involved in genome-wide regulation and evolution of adaptive traits.

Line 255: not described in the Results section!

Line 261: I think the conclusions in this paragraph are not supported by the data. Could the authors comment please?

Reviewer #3:

Remarks to the Author:

Review of “Transposable elements maintain genome-wide heterozygosity in inbred populations”
This study by De Kort et al reports association between heterozygosity estimated from RADseq-derived SNPs across populations of *Arabidopsis lyrata* and the presence of different transposable elements

(TEs) in the reference genome. The biological system is well established for its gradient of inbreeding across North America due to transitions towards selfing, which is here used to discuss possible contrasts between TE-flanking and "background" regions of the genome.

The manuscript is concisely written (sometimes to such an extent that superficial descriptions hamper proper understanding; see below) and, bringing fresh perspectives on long-held questions, appears typical of what is published in Nature Communication.

Although the introduction on selfing and purging offers valuable background (despite some seminal work by Willis on *Mimulus* being ignored), I found transposable elements being more superficially introduced. The supposedly 50x higher mutation rate around TEs should be clearly justified by literature (currently, based on loosely connected references). Also, available knowledge about TEs in *A. lyrata* and their evolution across populations shall be better introduced and used in the discussion (e.g. early insights from Gaut's lab a decade ago); also further justifying remaining gaps in our understanding that is here addressed.

A crucial issue that should be considerably clarified is the possible limits of the approach. First, that RADseq offers accurate genotyping (and particularly heterozygosity) should be validated with additional details on the here-used protocol (e.g. coverage) and in depth post hoc analyses. Second, and most importantly, TEs are known as mutagenic by themselves and polymorphic copies are therefore expected to segregate with the species, although only sites flanking TE copies inserted in the reference genome of *A. lyrata* are here under scrutiny. How was it taken into consideration that some TE loci may be absent and, even more pervasive, that quite many sites free of TEs in the reference can be populated by inserted copies across populations under scrutiny? How does such a reference-bias affect conclusions regarding SNPs flanking TEs? Finally, selection tests shall be spelled out and justified as conservative indirect estimates based e.g. on literature (the very limited neutral loci apparent on Fig.4a suggests non-conservative estimates). Generally, figures and their legends should be better linked to the core text and made comprehensive.

Reviewer 1:

This paper examines whether transposable elements play a role in “generating” genetic variation (heterozygosity) through increased mutation rates, thereby increasing adaptive potential. Specifically, the authors test whether highly inbred populations may benefit from this process (as a mechanism to compensate for the reduced overall genetic diversity due to inbreeding).

Using a RAD-seq approach, authors find evidence for elevated heterozygosity downstream of some TE superfamilies, and indications that this increased heterozygosity may be associated with functional genes through detection of signatures of selection (outlier detection) and gene ontology/enrichment analyses. In general, I think the paper addresses a **timely and original question**, takes an original approach and has a message that is of **broad interest**. However, it was not an easy read, and perhaps because of this, I have several major concerns, mostly related to the **methodology**.

1. Overall, I think the genomic analyses (i.e., the genotyping and TE identifications) were thorough. The only aspect I wonder about is whether the minimum read depth of 10 per called locus may have been on the low side? Authors may want to justify this choice in the paper. Related to this, please specify whether there was a minimum allele frequency required for a SNP to be called? Looking at the supplemental tables, SNPs are scored as 0 (hom) or 1 (het). Surely, for Bayescan, the actual allelic states were used? It would be good to provide these in a data repository, if this has not been done so already (in which case, please mention the link).

Response: While the minimum read depth of 10 per locus is far from unusual for RADseq projects (e.g. a minimum depth of 5x per locus in Kim et al. 2017 in *Nature Ecology & Evolution* doi.org/10.1038/s41559-017-0235-2, and minimum read depth of 10x in Yan et al. 2020 in *Nature Ecology & Evolution* doi.org/10.1038/s41559-019-1081-1), the average read depth in our study is significantly higher (European mean locus depth = 73.18; North American mean depth = 60.09). We now provide supporting figures displaying the range of read depths across the SNP dataset. The minimum allele frequency was set to 0.05, which reduces false SNP calls as well as rare deleterious SNPs that could impact the interpretation of our outlier detection methods (please see **comment 3** for more details on this issue). No missing data were allowed to avoid arbitrary imputation methods that could impact the accuracy of our data. For Bayescan, the actual allelic states were used; and we now add supporting tables with the vcf data containing this information (**Tables S3 and S4**). These vcf files were used for F_{IS} calculations, PCadapt, and Bayescan.

Methods New L372-374: “We limited the dataset to highly qualitative SNPs through maintaining (i) minimum allele frequencies of 0.05, (ii) no missing data, and (iii) an average read depth of 73.18 (European samples) and 60.09 (North American samples) per SNP (**Fig. S4**).”

Methods New L389: “(see Tables S3 and S4 for input files)”.

New Fig. S4 Distribution of mean read depth and mean genotype likelihood across (a) European samples and (b) North American samples for SNPs included in the study.

2. The statistical analyses were more difficult to follow, and I identified a few potentially major concerns:

Writing and presentation. Both the results and methods sections were hard to follow. In the results section, **subheadings** would be helpful to give some structure. Also, I would avoid too many almost literal stylistic repetitions of intro and methods, but rather **formulate the results as findings, with a brief interpretation in relation to the hypotheses**. For example: In support of *idea/hypothesis*, we found XX was related to YY.

Response: We now added subheadings, and re-formulated the results to facilitate their interpretation with respect to the corresponding hypotheses.

~~Old L141-145: “We tested whether (i) genome-wide heterozygosity at SNPs obtained through RAD-sequencing was particularly high downstream of TEs (Fig.1: Q1), and (ii) specific TE superfamilies are associated with elevated heterozygosity under intense inbreeding (Fig.1: Q2) using generalized mixed models. We focused on North American *A. lyrata* populations, which exhibit a mixed mating system and consequently represent an inbreeding gradient (Fig.1, Table S1).”~~

New L129-130: “In order to understand the relationship between TEs and nearby heterozygosity, we first explored the tendency of random SNPs obtained through restriction-site associated DNA sequences (RADseq) to occur near TEs.”

New L155-156: “Together, these findings suggest that TEs can impact downstream heterozygosity across the genome, in accordance with our hypothesis (Fig.1: Q1).”

~~Old L162-164: “We then tested the tendency of TEs to consistently (across independent evolutionary lineages) impact downstream vs. upstream adaptive evolution using outlier detection methods and gene enrichment analysis (Fig.1: Q3).”~~

New L195-197: “Together, these findings (Fig. 3, Fig. 4) support our hypothesis that TEs can provide evolutionary potential under inbreeding (Fig.1: Q2).”

New L198: “In line with our final hypothesis (Fig.1: Q3), we finally found that TE-specific signatures of selection (Fig.4C) were reproducible across evolutionary lineages.”

3. Bayescan – have authors ignored purifying selection? Especially in the results, but also the methods, I think the Bayescan approach should be described better. The results strongly rely on the Bayescan outlier detections. However, the resulting classes of loci (under balancing selection, under divergent selection & neutral) are very much taken as a given for making certain comparisons. It took me a while to figure out that these classes had been inferred with Bayescan (probably based on the alpha parameter, but this should be explained in the paper, neither the methods L351-354 mention Bayescan, nor do the figure captions). A major concern related to this is the interpretation of outlier loci with $\alpha < 0$ as under balancing selection. In my understanding (which is limited, so please rebut if I missed something obvious) it is not really possible with Bayescan to distinguish between signatures of balancing vs. purifying selection. Thus, unless the authors have taken analytical step enabling them to rule out purifying selection (if so, I don't think they have described it in their paper), their classification of loci with $\alpha < 0$ as being under balancing selection is misleading. I would personally expect strong signatures of purifying selection, as one would expect most mutations (including those due to TE activity) to be deleterious. If I am correct, this would somewhat weaken the overall author's interpretation that TE-associated mutations increase evolutionary potential, in which case the paper's general message should be toned down.

Response: We agree that purifying selection can give rise to signatures of apparent balancing selection when genetic variants are slightly deleterious. TE insertions have indeed been demonstrated to be subject to intense purifying selection due their overall deleterious impact on gene functioning (e.g. Baduel et al. 2021). Because mutations caused by TEs may have similar deleterious consequences, **we have now (i) explored the possibility of purifying selection, (ii) clarified the balancing selection methodology, and (iii) devoted part of the discussion to this issue.** Distinguishing purifying from balancing selection is a challenging task, particularly on low-density SNP sets where it is impossible to look at the shape of the allele frequency spectrum within genomic regions. However, a characteristic difference between signatures of balancing and purifying selection is the behavior of non-synonymous mutations (e.g. Charlesworth & Eyre-Walker 2008; Hernandez et al. 2019). Specifically, purifying selection more readily constrains non-synonymous alleles to low frequency than balancing selection, and can therefore be identified as low-frequency signatures of balancing selection. We now consider SNPs with negative alpha values as putative candidates of purifying selection if their alleles are rarer than expected under neutrality. More specifically, we consider SNPs with signatures of balancing selection as candidates of purifying selection if their minimum allele frequency (MAF) was (i) smaller than the MAF of neutral SNPs, and (ii) smaller for non-synonymous relative to synonymous codon usage. While SNPs near CACTA elements appear to be associated with some purifying selection, **we found low-frequency signatures of balancing selection to be rare**, particularly across the American populations (please see **New Fig. S3** below). This new insight, together with the **enrichment of functions typically involved in balancing selection (e.g. response to biotic stimuli and to hormones) (Kerwin et al. 2015, Wu et al. 2017), suggests that most of our SNPs with negative alpha values are linked to balancing rather than purifying selection.**

New Fig. S3. “**Minimum allele frequency (MAF) of non-synonymous vs. synonymous SNPs, for the American (left) and European (right) *A. lyrata* lineage.** MAF means are presented with standard errors per TE superfamily. TE superfamilies with too low sample sizes ($N=1$) are transparent (e.g. LAR in both panels). TE superfamilies associated with low frequency signatures of balancing selection are considered as candidates of purifying selection if their MAF is smaller than expected under neutrality, i.e. smaller than MAF of neutral SNPs, and smaller for non-synonymous than for synonymous codon usage. We find that neutral SNPs have the same MAF distribution in both lineages. For the American lineage, we further find that all TE superfamilies with $N>1$ are characterized by high frequency signatures of selection, indicating little purifying selection. For the European lineage, however, we identify signatures of purifying selection in several TE superfamilies (e.g. MuDR and Cacta), where MAF was smaller for SNPs with signatures of balancing selection than for neutral SNPs, and where non-synonymous SNPs in particular were associated with low MAF.”

Results New L172-182: “Because signatures of purifying selection can resemble those of balancing selection, we also explored the behavior of non-synonymous and synonymous substitutions in our set of putative balancing SNPs. Specifically, purifying selection more readily constrains non-synonymous substitutions to low frequency than balancing selection, and can therefore be identified as low-frequency signatures of balancing selection. While SNPs near CACTA elements appeared to be associated with low-frequency signatures of balancing selection, which could point to purifying selection acting on deleterious alleles, we found low-frequency signatures of balancing selection to be generally rare, particularly in the North American lineage (Fig. S3). Genomic regions with signatures of balancing selection were also significantly enriched for processes that have been frequently associated with balancing selection (e.g. response to biotic stimuli; Table S8), suggesting that our findings are not strongly impacted by purifying selection.”

Discussion New L325-335: “Our findings do not suggest a significant role for purifying selection and mutational meltdowns as drivers of genetic variation across the genome of self-fertilizing *A. lyrata*. While lethal TEs are rapidly purged from a population (Badel et al. 2021), slightly or conditionally deleterious alleles arising near TEs are more difficult to purge, and may reach higher frequencies resembling signatures of balancing selection (Fijarczyk & Wiestaw Babik 2015). The lack of low-frequency signatures of balancing selection (Fig. S3) suggests that purifying selection does not considerably contribute to genome-wide variation across American *A. lyrata* populations, with the enrichment of processes related to signaling and biotic responses particularly supporting a role for balancing selection (Table S8). In *A. thaliana*, genome-wide signatures of balancing selection are often associated with resistance and stress responses (Wu et al. 2017), and with temporally fluctuating selection pressures acting upon defense metabolism and signaling genes (Kerwin et al. 2015).”

Methods New L405-413: “To assess the role of purifying selection in producing negative alpha values, we explored the tendency of SNPs with negative alpha values to be associated with low-frequency non-synonymous variants. If alleles at these loci were rarer than expected under neutrality, we considered these SNPs candidates of purifying selection (Charlesworth & Eyre-Walker 2008;

Fijarczyk & Wiesław Babik 2015; Hernandez et al. 2019). Specifically, we considered SNPs with signatures of balancing selection as candidates of purifying selection if their minimum allele frequency (MAF) was (i) smaller than the MAF of neutral SNPs, and (ii) smaller for non-synonymous as compared to synonymous codon usage. Codon usage (4-fold and 0-fold positions in the *A. lyrata* reference genome) of individual SNPs was obtained using the *NewAnnotateRef.py* script (Williamson et al. 2014).”

New references:

- Charlesworth, J. & Eyre-Walker, A. The McDonald-Kreitman Test and Slightly Deleterious Mutations. *Mol. Biol. Evol.* **25**, 1007–1015 (2008).
- Fijarczyk, A. & Babik, W. Detecting balancing selection in genomes: limits and prospects. *Mol. Ecol.* **24**, 3529–3545 (2015).
- Hernandez, R. D. et al. Ultrarare variants drive substantial cis heritability of human gene expression. *Nat. Genet.* **51**, 1349–1355 (2019).
- Wu, Q. et al. Long-term balancing selection contributes to adaptation in Arabidopsis and its relatives. *Genome Biol.* **18**, 1–15 (2017).
- Kerwin, R. et al. Natural genetic variation in Arabidopsis thaliana defense metabolism genes modulates field fitness. *Elife* **2015**, 1–28 (2015).
- Baduel, P. et al. Genetic and environmental modulation of transposition shapes the evolutionary potential of Arabidopsis thaliana. *Genome Biol.* **22**, 1–26 (2021).
- Williamson RJ, Josephs EB, Platts AE, Hazzouri KM, Haudry A, Blanchette M, Wright SI. 2014. Evidence for widespread positive and negative selection in coding and conserved noncoding regions of *Capsella grandiflora*. *PLoS Genet.* 10(9):e1004622

4. Generalized linear mixed modelling. There are a couple of very confusing aspects to the modelling approach described: **a)** Relatedness seems to be a continuous variable but has been included in the model as a random effect. However, AFAIK, random effects are always categorical. So, it seems to me that relatedness should have been included as covariate. Despite the explanations in L318-320, I could not understand how it was calculated. What is a “sample” in this context? Relatedness to what? Is it simply the groupings arising from the PCA? **b)** I do not understand the logic of the competing background models, and I think it has not been explained how this works at all. Why is it better or more appropriate than using a single model with a binomial error term to explain the probability of heterozygosity (H) including fixed terms:

TEsuperfam, FIS, Selection, Poly(TEdist quadratic), TEsense, SNPsense and (a choice of) their interactions?

I am sure there is good reasons for the taken approach, but the reader needs much more guidance. It is currently impossible to judge whether the taken modelling approach is appropriate or not.

Response: We confirm that the random effect “**Relatedness**” consists of categorical groups obtained by PCadapt. This was explained in the Methods, but we now also mention it in the Results section and in the caption of a new table so as to facilitate the interpretation of the results (**New Table 1**). Our **modeling choices** were based on two criteria: (i) providing a logical stepwise approach matching the hypotheses, and (ii) avoiding model convergence issues and minimizing model complexity. We thus first identified those TE superfamilies that significantly impact heterozygosity under inbreeding (cfr. our main hypothesis), which considerably reduced the degrees of freedom in subsequent models from 13 TE superfamilies (x interaction terms) down to one grouping variable discriminating between relevant TE superfamilies (group1) and background TE superfamilies (group2). We now also **simplify most of the models**, rendering them more **compatible with the hypotheses**. Specifically, we removed the “TE_Sense” variable from our first model, since it was not connected to our hypotheses and did not significantly impact heterozygosity. In addition, because distance to nearest TE did not contribute to Model 1, it was removed from subsequent models. Note that **these new adjustments have not changed our key findings**. One minor result that was difficult to explain in our previous version (i.e. the behavior of heterozygosity with distance to nearest TE), however, was not

significant in these revised models. We changed the discussion on the distance effect accordingly (**New Lxxx**, see below). Finally, the use of **competing models** (H_{TE} vs. H_0) served to avoid complicated three-way interactions: “existing two-way interactions \times TE group” and to facilitate the interpretation of our findings (see adjusted Methods section below for more details). To emphasize the relations between our models and our research questions, we also **added a table** that summarizes the models and their corresponding hypotheses. **We hope that the rationale behind our approach is now clearer to the readers.**

Methods New L416-428: “We followed a stepwise modeling approach that allowed testing our hypotheses consecutively without running complex models with many parameters. We first ran an explorative mixed model (Model 0, **Table 1**) to: (i) identify TE superfamilies with elevated levels of heterozygosity under inbreeding, and (ii) reduce data complexity in subsequent models through replacing the TE superfamily variable ($N=13$ levels) by a simple TE grouping variable (“**TE Effect**”, $N=2$ levels, **Box 1**). This new grouping variable distinguishes between TE superfamilies that significantly impact heterozygosity under inbreeding (H_{TE}) and all other TEs (H_0). To test whether this effect was exclusively associated with high inbreeding levels, the explorative model was run on a subset of the samples characterized by predominant self-fertilization and with inbreeding coefficients exceeding 0.6 (Model 0a), and compared to a model with subset of the samples characterized by predominant outcrossing and with inbreeding coefficients below 0.6 (Model 0b).”

Methods New L432-439: “Model 1 (shown in **Table 1**) was built to test our first hypothesis (Q1 in **Fig. 1**), i.e. how does the position of a SNP relative to its nearest TE affect its heterozygosity? To this end, we explored the interactive effect between “Distance” and the orientation of a SNP relative to its nearest TE (“SNP_sense”: downstream vs. upstream of TE vs. within TEs) on the sensitivity of a SNP to changes in heterozygosity. “Distance” was included to explore the tendency of SNPs to be heterozygous with decreasing distance to the nearest TE (linear) or at intermediate distance (quadratic). We hypothesized the relative position of a SNP to have a larger effect on H_{TE} than on H_0 , and therefore also included “TE Effect” in interaction with “Distance” and “SNP_sense” in Model 1 (**Table 1**).”

Methods New L440-449: “Model 2a (**Table 1**) relates to our second hypothesis (Q2 in **Fig. 1**). Specifically, we tested whether nearby heterozygosity (H_{TE}) caused deviations from the expected negative relationship between heterozygosity and inbreeding, and whether this deviation was associated with signatures of selection. The variable “Selection” indicated whether a SNP was an outlier of balancing selection or of divergent selection rather than a neutral SNP (three levels). To test whether the patterns observed for Model 2a are unique to the TE superfamilies identified in Model 1, we compared the Model 2a outcomes to a competing background model (Model 2b) ran on all other TE superfamilies. We expected the inbreeding coefficient to show a stronger correlation with H_0 (Model 2b) than with H_{TE} (Model 2a, cfr. Q2 in **Fig. 1**). We preferred competing models over one all-inclusive model to avoid complex three-way interactions (fictional Model 2: $H \sim FIS \times Selection \times TE\ Effect + Sense \times Selection \times TE\ Effect$).”

Methods New L454-456: “Model 3 (**Table 1**) was built to test our third hypothesis (Q3 in **Fig. 1**) assessing the TE effect (impact of TE superfamilies on heterozygosity under inbreeding) on an evolutionary and spatially independent *A. lyrata* lineage that is characterized by an outcrossing mating system.”

New L569-577: “**Table 1.** List of generalized mixed models used to test our three main hypotheses. The probability of a locus to be heterozygous was consistently modeled with the random effect “relatedness”, a categorical variable correcting for genetic relatedness among samples as revealed by a multivariate PCadapt analysis. TE superfamilies with significant vs. no impact on heterozygosity under inbreeding (Model 0) were grouped into the binary grouping variable “TE_Effect”. We tested both a linear and quadratic effect of the “Distance” between a SNP and its nearest TE on its heterozygosity. See **box 1** for terminology with regard to heterozygosity (H , H_{TE} , H_0), and **Fig. 1** for an overview of the three corresponding hypotheses”.

Model		Hypothesis (Fig. 1)
-------	--	---------------------

0a ($F_{IS} > 0.6$)	$H \sim TE \text{ Superfamily:Sense} + \text{Distance}$	Explorative
0b ($F_{IS} < 0.6$)	$H \sim TE \text{ Superfamily:Sense} + \text{Distance}$	Explorative
1	$H \sim \text{Distance} \times TE \text{ Effect} + TE \text{ Effect} \times \text{Sense} + \text{Distance} \times \text{Sense}$	Q1
2a	$H_{TE} \sim FIS \times \text{Selection} + \text{Sense} \times \text{Selection}$	Q2
2b	$H_0 \sim FIS \times \text{Selection} + \text{Sense} \times \text{Selection}$	Q2
3	$H (\text{Europe}) \sim TE \text{ Effect} + \text{Sense}$	Q3

Models used in previous version (to clarify the changes we made to the revised version):

	Hypothesis
$H \sim TE \text{ Superfamily} + \text{Distance} + TE_sense + SNP_sense (F_{IS} > 0.6)$	Explorative + Q1
$H_{TE} \sim FIS + \text{Distance} \times SNP_sense$	unclear
$H_0 \sim FIS + \text{Distance} \times SNP_sense$	unclear
$H_{TE} \sim FIS \times \text{Selection} + \text{Distance} \times \text{Selection}$	Q2
$H_0 \sim FIS \times \text{Selection} + \text{Distance} \times \text{Selection}$	Q2
$H \sim TE \text{ effect} + \text{Distance} \times SNP_sense$	Q3

Results New L149-151: “While *Copia* and *Harbinger* elements were associated with elevated levels of heterozygosity particularly under inbreeding, *LINE* elements had consistently high heterozygosity irrespective of inbreeding levels (**Fig. 3A,B**). The TE effect thus was limited to *Copia* and *Harbinger* elements.”

Discussion New L232-240: “Interestingly, however, while mutation rates have been shown to decrease gradually with genomic distance from TEs (Wicker et al. 2016), we don’t observe equivalent effects on heterozygosity. We suspect this finding to result from TE insertion preference. Specifically, TEs tend to insert (i) near functional genes, where flanking mutations in particular can provide fitness benefits to the host (Quadrona et al. 2019; Baduel et al. 2021), and (ii) away from vital genes, where flanking or more distal mutations can be harmful or beneficial to organismal functioning depending on the distance between the TE and the gene. As a result, the genome-wide distribution of proximity effects of TEs on heterozygosity can vary with gene function and essentiality.”

5. FIS. The authors use FIS as an estimation of inbreeding levels, and it is reassuring that the highest values correspond to populations known to display high selfing rates. Still, I disagree that we truly have a gradient here. In line with what we know from earlier work, most populations are either highly outcrossing, or highly selfing. The FIS values reflect this, as data points are really sparse between $FIS=0.5$ and $FIS=0.9$. The patterns in Fig 4 D and E remain extremely interesting regardless, with clearly increased heterozygosity at all FIS levels for loci under purifying (or balancing) selection.

Response: We now explicitly test our TE effect in the self-fertilizing vs. outcrossing samples (Models 0a and 0b, please see reply to **comment 4**), and added a panel (**Fig. 3B**) showing that heterozygosity near *Copia* and *Harbinger* elements strongly differs between the highly inbred vs. outcrossing American populations. This could suggest a role for mating system as a potential driver of heterozygosity-TE relationships. To explore this, we now disentangle the potential role of mating system strategy from other processes shaping inbreeding coefficients through subsetting the European individuals (all characterized by a predominantly outcrossing mating system) according to their inbreeding coefficients. Specifically, we divided the European dataset into individuals from populations that likely faced stronger demographic bottleneck across glacial periods ($F_{IS} > -0.3$) vs. individuals with $F_{IS} < -0.3$) (Mattila et al. 2017). We found a marginal TE Effect on heterozygosity in Europe (see Table S9 and **New Fig. S1**), suggesting that demographic processes rather than mating strategy govern TE-heterozygosity relationships.

Methods New L422-426: “To test whether this effect was exclusively associated with high inbreeding levels, the explorative model was run on a subset of the samples characterized by predominant self-fertilization and with inbreeding coefficients exceeding 0.6 (Model 0a), and compared to a model on a subset of the samples characterized by predominant outcrossing and with inbreeding coefficients below 0.6 (Model 0b).”.

Methods New L468-473: “To disentangle mating strategy effects from demographic effects on the inbreeding coefficient, we explored this expectation on two subsets that share the same outcrossing mating system but differ in inbreeding coefficients ($F_{IS} > -0.3$ vs. $F_{IS} < -0.3$), presumably due to strong demographic bottlenecks experienced by the most inbred individuals experience by the most inbred European individuals (Mattila et al. 2017). If mating system strategy is the driver of TE effects, then $H_0 = H_{TE}$ for both subsets. If demography drives TE effects, then $H_0 = H_{TE}$ only for the subset with $F_{IS} < -0.3$.”

Results New L209-212: “The TE effect was, however, marginally present in outcrossing European populations with $F_{IS} > -0.3$ (Fig. S1, Table S9), suggesting that demographic processes rather than mating system strategy drive the TE effect on heterozygosity under intense inbreeding.”

New Fig. 3. “Fig.3. Relationships between heterozygosity and TE characteristics. Heterozygosity in inbred individuals (genome-wide $F_{IS} > 0.6$) was generally low but was elevated near Copia, Harbinger and LINE elements (A). While Copia and Harbinger elements can maintain heterozygosity particularly under inbreeding, only LINE elements are also associated with increased heterozygosity in outcrossing populations (genome-wide $F_{IS} < 0.6$) (B). We therefore considered the Copia and Harbinger elements to contribute to the “TE Effect” (box 1, Table 1). As the CACTA superfamily was associated with intermediate levels of heterozygosity in the American *A. lyrata* lineage, we used CACTA as the reference group (intercept) for statistical comparisons with other TE superfamilies in the mixed model output of Table S6. Heterozygosity also depended upon the signature of selection (C and D), with balancing selection typically increasing heterozygosity, particularly downstream of TEs. Background heterozygosity (H_0) refers to heterozygosity near all TE superfamilies but Copia and Harbinger (see Box 1). Significance of factor level estimates are provided in Table S6.”

New Fig. S1: “Probability of a SNP being heterozygous near Copia and Harbinger elements vs. all other transposable elements. The results (probability estimates obtained through mixed model 3) are presented separately for *A. lyrata* plants characterized by relatively low (left panel) vs. high (right panel) genome-wide inbreeding coefficients. See Table S9 for associated summary statistics.”

New reference:

Mattila TM, Tyrmi J, Pyhäjärvi T, Savolainen O. 2017. Genome-wide analysis of colonization history and concomitant selection in *Arabidopsis lyrata*. *Molecular Biology and Evolution* 34: 2665-2677.

6. Tests of significance. Authors mainly report F values, but they seemingly mention only one degree of freedom (I assume the residual df are not mentioned). For example - L136 F13=4.98**. Meaningless if we do not know the residual df. Moreover, the * are not explained (I assume ** means $p < 0.01$). In my understanding, F values indicate whether fixed effects are significant or not. However, if such fixed effects have multiple levels, they do not inform about differences between the factor-levels, for which post hoc comparisons are required. It only became clear after looking at Table S5 that in reality the significance of factor level estimates are evaluated (z-tests).

Response: The subscript with our F values do not refer to degrees of freedom but to number of parameters, which is provided by the ANOVA tables for generalized models (as opposed to general models, where degrees of freedom are provided). We now removed these subscripts to avoid confusion, and mention in corresponding figure captions that significance of factor level estimates are provided in the corresponding supplementary tables.

Fig. 3 New L548: “Significance of factor level estimates are provided in Table S6.”

Fig. 4 New L557: “Significance of factor level estimates are provided in Table S6.”

7. Related to the previous point, I struggled to understand what the intercept represented in the models. For example, there are 13 TE superfamilies in figure 3A, and 13 factor levels listed in Table S5, plus an intercept (i.e., 14 levels). It is probably my ignorance, and has a very simple explanation, but it would be good to include this explanation in the paper. Likewise, I struggled to understand how a binomial/binary Y can be zero-inflated.

Response: The intercept represented the first TE superfamily in alphabetical order (i.e. CACTA). We kept this arbitrary choice by the statistical software because the CACTA superfamily was characterized by intermediate heterozygosity levels that represent genome-wide heterozygosity (please see Fig. 3A). We clarified this in our Methods section and in the caption of Fig.3. Given the large number of factor levels, a full statistical contrast of all factor levels would be difficult to interpret and not offer more clarity. Note that our original model also erroneously included TEs with unknown superfamily (SuperfamilyTE_NA); we have now excluded this group from our dataset

(hence the total number of TE superfamilies is now 13 including CACTA). We agree that binomial data cannot be statistically zero-inflated, but rather used this term to emphasize the high proportion of zeros in our dataset (because most SNPs are homozygous, value “0”). To avoid confusion, we have removed the zero-inflation statement from the text (**New Lxxx**).

Fig. 3 New L542-545: “The CACTA superfamily was associated with intermediate levels of heterozygosity in the American *A. lyrata* lineage, and therefore taken as a reference group in the mixed model output of **Table S6**.”

8. TE superfamily. Could an individual SNP be affected by more than one TE superfamily? I think the models assume that SNP heterozygosity can only be affected by its closest TE, but how realistic is this? Could there be interactive effects of TEs?

Response: Many TEs are indeed clustered into TE islands, and relationships between SNPs and their closest TE may represent an indirect relationship with another, more distant, TE. We now discuss the possibility of interacting TE effects in our manuscript, while emphasizing that our findings represent the genome-wide impact of TEs on genetic variation under inbreeding, irrespective of TE clustering behavior. TE interactions might indeed be expected to hide TE effects of individual TE superfamilies rather than generate false positive findings. We therefore believe our results are conservative in interpreting the effect of individual TE superfamilies.

Discussion New L344-349: “Second, many TEs cluster into TE islands (Schrader et al. 2014, De Kort et al. 2021), corresponding to the idea that functional genomic regions are frequently revisited by TEs (Baduel et al. 2021). In TE islands, the associations between heterozygosity at specific genetic variants and their nearest TE may be impacted by a slightly more distant TE. Such TE interactions may have reduced our ability to detect significant effects of individual TEs.”

New reference:

De Kort H, Panis B, Deforce D, Van Nieuwerburgh F, Honnay O. 2020. Ecological divergence of wild strawberry DNA methylation patterns at distinct spatial scales. *Molecular Ecology* 29: 4871–4881.

I also have a few conceptual concerns:

9. Inbreeding depression/purging as a strategy. In my view, purging is not a strategy, it just happens, as a consequence of inbreeding. L45 – deleterious alleles may be purged, but not complete loci. In the discussion, there is a misconception that repeated selfing increases the risk of inbreeding depression (L214-216). Given purging, the opposite is true.

Response: We agree and rephrased.

Introduction New L45: “Purging of alleles associated with inbreeding depression therefore is thought to contribute to the success of self-fertilizing species.”

Introduction New L48: “While purging can be an effective strategy to safeguard ...” → “While purging can safeguard...”

Discussion New L244: “Extreme inbreeding levels typically increase the risk for inbreeding depression,...”.

10. The motivation for this study is understanding “the evolutionary success of selfers”. While I completely agree that it is intriguing that selfers manage to occupy a wide range of habitats, they might not necessarily achieve this through adaptation. The alternative view – that selfers are evolutionary dead-ends, but may evolve repeatedly due to short term advantages – also requires attention. Under this scenario, selfers do not need to adapt per se. They may arise pre-adapted to

the prevailing conditions (from a locally adapted outcrossing ancestor), or many formed selfing lineages may arise, after which selective filtering allows establishment of those lineages that are best-adapted to the prevailing conditions. Either way, such lineages may eventually go extinct when those conditions change due to limited adaptive potential. **It would be good to discuss whether the observed patterns of increased heterozygosity in highly inbred individuals could actually reflect mutational meltdown rather than increased adaptive potential.** This would at least explain the finding of many loci being under purifying selection (interpreted by the authors as balancing selection). The work by Emma Goldberg et al (Science 2010) comes to mind. In any case, I would play down L191-194 a bit.

Response: We thank the referee for pointing us into the direction of purifying selection and mutational meltdowns. As detailed in our reply to **comment 3** (on purifying selection) and **comment 5** (selfers vs outcrossers), we now explore (i) the behavior of low-frequency non-synonymous mutations (which are more likely to be under purifying selection than more common non-synonymous mutations), and (ii) the TE effect across the outcrossing populations in Europe (no mating system variation but a partial F_{IS} gradient ranging from -0.237 to -0.637). We found limited signatures of purifying selection (please see **comment 3** for details). In addition, we found that both in the American and European lineage, Copia and Harbinger elements were associated with increased heterozygosity where inbreeding levels were relatively high (New **Fig. 3A and 3B**), suggesting that (i) the TE effect is linked to levels of homozygosity (which in turn are influenced by demographic processes) rather than to the mating system, and (ii) signatures of balancing selection unlikely reflect an ongoing mutational meltdown. Note that while these additional findings suggest that the TE effect associated with inbreeding also arises in outcrossing species, it may still partially explain the success of self-fertilizing species (at least under the assumption that mutational meltdown is not an issue), through facilitating evolution under genome-wide genetic depletion. The idea that our signatures of selection reflect pre-adaptation and that self-fertilizing *A. lyrata* populations are not capable of evolving is interesting, but not in line with our finding of elevated genetic variation in stress-responsive genes, which suggest adaptive potential (see **New L195-197**). Finally, we now refer to a study from Wright et al. (2002) specifically addressing the issue of mating system on rates and patterns of molecular evolution in inbred and outbred *Arabidopsis*, where no indications of mutational meltdown associated with a long history of self-fertilization could be identified. We also recognize and discuss the possibility of mutational meltdown in our paper.

Introduction New L31-33: “*Arabidopsis thaliana*, for instance, a self-compatible species with extreme inbreeding levels across its natural range, does not show widespread signatures of increased extinction risk but instead exhibits significant adaptive potential (Wright et al. 2002; Atwell et al. 2010; Platt et al. 2010).”

Discussion New L254-259: “Interestingly, these findings are compatible with earlier studies showing that self-fertilizing *A. lyrata* populations do not exhibit reduced fitness as compared to outcrossing populations (Willi 2012; Joschinski et al. 2015). The potential existence of TE-related molecular strategies maintaining heterozygosity at fitness-related genomic regions represents a compelling line of research that can explain the lack of inbreeding depression under intense inbreeding.”

Discussion New L317-324: “More broadly, this theory could explain the success of self-fertilizing populations across the globe. Importantly, this does not refute the idea that self-fertilizing populations frequently are evolutionary dead-ends arising from continuous drift-induced accumulations of deleterious mutations (Stebbins 1957; Takebayashi & Orrell 2001; but see Igic & Busch 2013). Specifically, while rates of extinction have been found to be significantly higher in self-pollinating as compared to outbreeding taxa (Goldberg et al. 2010; Abu Awad & Billiard 2017), self-pollinating taxa that do persist through time may do so through evolving strategies counteracting inbreeding depression.”

New references

- Wright, S. I., Lauga, B. & Charlesworth, D. Rates and Patterns of Molecular Evolution in Inbred and Outbred Arabidopsis. *Mol. Biol. Evol.* **19**, 1407–1420 (2002)
- Takebayashi, N. & Morrell, P. L. Is self-fertilization an evolutionary dead end? Revisiting an old hypothesis with genetic theories and a macroevolutionary approach. *Am. J. Bot.* **88**, 1143–1150 (2001).
- Stebbins, G. L. Self Fertilization and Population Variability in the Higher Plants. *Am. Nat.* **91**, 337–354 (1957).
- Igic, B. & Busch, J. W. Is self-fertilization an evolutionary dead end? *New Phytol.* **198**, 386–397 (2013).
- Goldberg, E. E. *et al.* Species selection maintains self-incompatibility. *Science.* **330**, 493–495 (2010).
- Abu Awad, D. & Billiard, S. The double edged sword: The demographic consequences of the evolution of self-fertilization. *Evolution.* **71**, 1178–1190 (2017).
- Willi, Y. *Mutational meltdown in selfing Arabidopsis lyrata.* *Evolution.* **67**, 806–815 (2013).
- Joschinski, J., van Kleunen, M. & Stift, M. Costs associated with the evolution of selfing in North American populations of *Arabidopsis lyrata*? *Evol. Ecol.* **29**, 749–764 (2015).

Minor remarks

11. Fig 1 – Are any of the European samples polyploid? If so, does this affect the outcome in any way? Response: Although *A. l. petraea* is known to vary in ploidy across its range, only confirmed diploid populations were included (Schmickl *et al.* 2008) (Barbara Mable, pers. Comm).

Methods New L359: “Only confirmed diploid populations were used (Schmickl *et al.* 2008).”

New reference:

Schmickl R, Jørgensen MH, Brysting AK, Koch MA. 2008. Phylogeographic implications for the north American boreal-arctic *Arabidopsis lyrata* complex. *Plant Ecol Divers.*1(2):245–54.

12. L229 – balancing SNPs?? Do you mean SNPs with signatures of balancing selection (or rather purifying selection)? Response: Because our new data point to balancing rather than purifying selection as the predominant driver of signatures of balancing selection (please see reply to **comment 3**), we rephrased to “SNPs with signatures of balancing selection” (**New L241 and L409**).

13. L299 – please indicate where you obtained this reference genome, as there are multiple versions circulating. Response: **Methods New L381-382:** “... to annotate TEs in the *A. lyrata* assembly also used for RAD sequencing (Phytozome v11)...”

14. L302 – “only those” – only those SNPs or only those RAD loci? Response: only SNPs (see **New L371**).

15. L232-239 This paragraph feels like a loose end not tied in with any findings. Should it be integrated into the next paragraph? Response: We agree and now integrated our findings with respect to adaptive potential in this paragraph (**New L267-284**).

Reviewer 2:

In this manuscript by De Kort and colleagues the authors have reanalyzed RADseq data from 91 individuals of *Arabidopsis lyrata* (from US and EU, a total of around 5900 SNPs) with contrasted breeding systems, in order to test the potential role of transposable elements (TEs) in maintaining genetic variation in inbreeding conditions. While **the biological hypothesis is very novel and promising**, I see maybe **one major drawback** in this study. The study investigates the proximity of SNPs with given TEs based on the *lyrata* genome annotated with REPET.

16. It is well documented that a plethora of TE insertion polymorphisms (TIPS) are found in plant populations, even in narrow geographical areas (see in Nature Comm 2019 the works by Quadrana et al for *Arabidopsis thaliana* <https://doi.org/10.1038/s41467-019-11385-5> and Carpentier et al, for *Oryza sativa* <https://doi.org/10.1038/s41467-018-07974-5> for instance). In this context, how can the authors estimate the precise location of the TE families they investigate?

Response: We realize that TE insertions and other indels in genomes other than the reference genome likely are abundant across the sampling range, so we have an incomplete image of the true extent that TEs might enhance nearby heterozygosity. Nevertheless, several arguments justify the use of reference genome positions across our dataset for the purpose of our study. First, recent TE insertions in between our nearest neighbor SNP-TE pairs are expected to reduce the power of our analyses rather than generate false positive results. Second, our hypotheses focus on the impact of TE superfamilies on genome-wide heterozygosity and the orientation of SNPs relative to their nearest TE (downstream vs upstream). The precise positions of the SNPs and their nearest TE therefore is of lesser importance for most of our analyses. Third, as the reference genome is American, the largest impact of non-reference TE insertions could be expected in the European dataset, yet we found repeatable patterns for the American and European dataset. Finally, a recent study by Baduel et al. (2021) in *A. thaliana* demonstrated that the majority of TIPS are found at a very low frequency as a result of intense purifying selection acting upon deleterious TE insertions, unless they inserted into functional genomic regions. Such functional regions are frequently revisited by TEs, and explain the existence of TE islands near functional genes. We refer to our reply to **comment 8** on the potential effect of TE interactions on heterozygosity. We also added some lines to our manuscript to show that we recognize the ubiquity of TIPS and that they may have created noise in our findings.

Discussion New L342-349: “Recent findings in *A. thaliana* confirm that non-reference TE insertions appear at low frequency due to their deleterious nature, unless they integrate in functional genomic regions where natural selection facilitates the spread of TEs (Baduel et al. 2021). Second, many TEs cluster into TE islands (Schrader et al. 2014, De Kort et al. 2021), corresponding to the idea that functional genomic regions are frequently revisited by TEs (Baduel et al. 2021). In TE islands, the associations between heterozygosity at specific genetic variants and their nearest TE may be impacted by another nearby TE. Such TE interactions may have downplayed some of our findings.”

Methods New L384-386: “Through focusing on nearest neighbors, the probability of non-reference TE insertions (which typically arise at low frequency and randomly across samples; Baduel et al. 2021) even closer to our SNPs is strongly reduced.”

Reference:

- Baduel, P. et al. Genetic and environmental modulation of transposition shapes the evolutionary potential of *Arabidopsis thaliana*. *Genome Biol.* **22**, 1–26 (2021).

Minor comments:

17. Line 209: Overstatement: The simultaneous enrichment of biological regulation and stress-responsiveness found here suggests that TEs are involved in genome-wide regulation and evolution of adaptive traits. **Response:** We removed this sentence because it is redundant given the sentence before (**New L277-278**).

18. Line 255: not described in the Results section! **Response:** Indeed; we now described these compelling findings in the results section. **Results New L187-189:** “Interestingly, SNPs downstream of (i) MuDR TEs were enriched for processes involved in photosynthesis and growth, (ii) Helitron were typically associated with abiotic stresses (heat, light, drought), and (iii) MITEs with wounding and disease (Table S8).”

19. Line 261: I think the conclusions in this paragraph are not supported by the data. Could the authors comment please? Response: The relationship of Copia and Harbinger elements with inbreeding suggests their involvement in specific demographic scenarios. In addition, the relationship of (i) MuDR TEs with photosynthesis and growth, (ii) Helitron with abiotic stresses (heat, light, drought), and (iii) MITEs with wounding and disease, suggests their involvement in specific environmental scenarios. We nevertheless rephrased so as to make clear that further research is required to consolidate these findings.

New L306-307: *“While calling for further research, our results collectively suggest that each TE superfamily can facilitate evolution in particular environmental and demographic situations. Specifically,”*

Reviewer 3:

Review of “Transposable elements maintain genome-wide heterozygosity in inbred populations”
This study by De Kort et al reports association between heterozygosity estimated from RADseq-derived SNPs across populations of *Arabidopsis lyrata* and the presence of different transposable elements (TEs) in the reference genome. The biological system is well established for its gradient of inbreeding across North America due to transitions towards selfing, which is here used to discuss possible contrasts between TE-flanking and “background” regions of the genome. The manuscript is concisely written (sometime to such an extent that superficial descriptions hamper proper understanding; see below) and, **bringing fresh perspectives on long-held questions, appears typical of what is published in Nature Communication.**

20. Although the introduction on selfing and purging offers valuable background (despite some seminal work by Willis on *Mimulus* being ignored), I found transposable elements being more superficially introduced. The supposedly 50x higher mutation rate around TEs should be clearly justified by literature (currently, based on loosely connected references). Also, available knowledge about TEs in *A. lyrata* and their evolution across populations shall be better introduced and used in the discussion (e.g. early insights from Gaut’s lab a decade ago); also further justifying remaining gaps in our understanding that is here addressed.

Response: We completed our selfing and purging introduction with a key study by Willis (1999). We also toned down the mutation rate statement, as 50fold increases are rare. Increases of 10x and 15x are more common and explicitly shown in Wicker et al. 2016 and Habig et al. 2021. We also clarified the role of transposable elements in evolution, with an emphasis on what is known in *A. lyrata*. Indeed, Gaut’s lab represents much of what is known about the demographic and evolutionary history in *A. lyrata* and the role of TEs herein. We thus added two more references involving the work of Gaut’s lab.

New L38: *“(Lynch et al. 1995; Willis 1999; Coron et al. 2013; Kyriazis et al. 2020)”*

New L62-65: *“Transposable elements (TEs) can rapidly generate genetic variation through increasing mutation rates of upstream and downstream flanking genomic regions (Schrader et al. 2014; Lu et al. 2017; Schrader and Schmitz 2019; Habig et al. 2021); several studies have observed increases over 10x the background mutation rate (Wicker et al. 2016; Habig et al. 2021).”*

New L102-112: *“In the mixed mating species *Arabidopsis lyrata*, self-fertilizing populations are characterized by increased TE copy numbers and higher TE frequencies than outcrossing populations, aligning with models of neutral evolution and ectopic recombination (Bonchev & Willi 2018). More specifically, the higher levels of sequence homozygosity in self-fertilizing populations reduces the rate at which nonhomologous sequences recombine, thereby generating harmful chromosomal rearrangements. As a consequence, self-fertilization can facilitate genome-wide TE accumulation,*

potentially providing considerable opportunities for evolution in genomic sequences flanking TEs. However, while inbreeding may affect TE dynamics and evolution in *A. lyrata* (Lockton et al. 2010; Bonchev & Willi 2018; Lockton et al. 2018), the impact of inbreeding on genetic variation and evolution surrounding TEs has never been explicitly tested.”

New references:

- Willis JH. 1999. The role of genes of large effect on inbreeding depression in *Mimulus guttatus*. *Evolution* 53: 1678-1691
- Bonchev G & Willi Y. 2018. Accumulation of transposable elements in selfing populations of *Arabidopsis lyrata* supports the ectopic recombination model of transposon evolution. *New Phytol.* 219: 767–778.
- Lockton S, Ross-Ibarra J & Gaut BS. 2008. Demography and weak selection drive patterns of transposable element diversity in natural populations of *Arabidopsis lyrata*. *Proc. Natl. Acad. Sci. U. S. A.* 105: 13965–13970.
- Lockton S & Gaut BS. 2010. The evolution of transposable elements in natural populations of self-fertilizing *Arabidopsis thaliana* and its outcrossing relative *Arabidopsis lyrata*. *BMC Evol. Biol.* 10: 10.

21. A crucial issue that should be considerably clarified is the possible limits of the approach. First, that RADseq offers accurate genotyping (and particularly heterozygosity) should be validated with additional details on the here-used protocol (e.g. coverage) and in depth post hoc analyses.

Response: While the minimum read depth of 10 per locus is far from unusual for RADseq projects (e.g. a minimum depth of 5x per locus in Kim et al. 2017 in *Nature Ecology & Evolution* doi.org/10.1038/s41559-017-0235-2, and minimum read depth of 10x in Yan et al. 2020 in *Nature Ecology & Evolution* doi.org/10.1038/s41559-019-1081-1), the average read depth is much higher (73.18 in European samples and 60.09 in North American samples). We now provide supporting files demonstrating the range of read depths across the SNP dataset. The minimum allele frequency was set to 0.05, which further reduced false SNP calls as well as rare deleterious SNPs that could impact the interpretation of our outlier detection methods. No missing data were allowed to avoid arbitrary imputation methods that could impact the accuracy of our data.

We are not sure which post hoc analyses should be conducted to validate the approach. Given specific suggestions of such analyses, we would be happy to implement additional validation of the dataset if still deemed necessary. However, we do note that this RAD-seq dataset has been used successfully in several peer-reviewed scientific manuscripts (Buckley et al. 2018, Buckley et al. 2019).

References:

- Buckley J, Holub EB, Koch MA, Vergeer P, Mable BK. 2018. Restriction associated DNA-genotyping at multiple spatial scales in *Arabidopsis lyrata* reveals signatures of pathogen-mediated selection. *BMC Genomics* 19:1–21
- Buckley J, Daly R, Cobbold CA, Burgess K, Mable BK. 2019. Changing environments and genetic variation: natural variation in inbreeding does not compromise short-term physiological responses. *Proc. R. Soc. B Biol. Sci.* 286:20192109.

Methods New L372-374: “We limited the dataset to highly qualitative SNPs through maintaining (i) minimum allele frequencies of 0.05, (ii) no missing data, and (iii) an average read depth of 73.18 (European samples) and 60.09 (North American samples) per SNP (Fig. S4).”

Fig. S4 Distribution of mean read depth and mean genotype likelihood across (a) European samples and (b) North American samples for SNPs included in the study.

22. Second, and most importantly, TEs are known as mutagenic by themselves and polymorphic copies are therefore expected to segregate with the species, although only sites flanking TE copies inserted in the reference genome of *A. lyrata* are here under scrutiny. How was it taken into consideration that some TE loci may be absent and, even more pervasive, that quite many sites free of TEs in the reference can be populated by inserted copies across populations under scrutiny? How does such a reference-bias affect conclusions regarding SNPs flanking TEs?

Response: We fully agree that variable TE insertions (TIPs) are likely to populate our dataset. Importantly, however, a recent study in *Arabidopsis thaliana* (Baduel et al. 2021) highlights that the majority of recent TE insertions are deleterious and rare, in agreement with purifying selection against TEs that could insert between our SNPs and their nearest reference TE. As a result, it is unlikely that rare and randomly distributed non-reference TEs would have notable effects on our conclusions. Even when this is the case, it would likely just reduce the power of our analyses for detecting effects of TE superfamilies. As for reference TEs being absent across the dataset, we assume limited impacts on our result for two reasons: (1) common TE insertions typically re-visit genomic regions already occupied by TEs because genetic variation in these genomic regions provides adaptive potential (Baduel et al. 2021). This is reflected by our results because we found predictable patterns of natural selection between independent evolutionary lineages, suggesting that reference TEs involved in adaptive evolution are common across the dataset; (2) similar to non-reference TEs, reference TEs that are absent across the range are expected to be rare, and to add noise to our models rather than generate false positive findings. We discuss this issue in more depth (see also our reply to **comment 16**).

Discussion New L342-349: “Recent findings in *A. thaliana* confirm that non-reference TE insertions appear at low frequency due to their deleterious nature, unless they integrate in functional genomic regions where natural selection facilitates the spread of TEs (Baduel et al. 2021). Second, many TEs cluster into TE islands (Schrader et al. 2014, De Kort et al. 2021), corresponding to the idea that functional genomic regions are frequently revisited by TEs (Baduel et al. 2021). In TE islands, the associations between heterozygosity at specific genetic variants and their nearest TE may be impacted by another nearby TE. Such TE interactions may have reduced our ability to detect significant effects of individual TEs.”

Methods New L384-386: “Through focusing on nearest neighbors, the probability of non-reference TE insertions (which typically arise at low frequency and randomly across samples; Baduel et al. 2021) even closer to our SNPs is strongly reduced.”

Reference:

Baduel, P. et al. Genetic and environmental modulation of transposition shapes the evolutionary potential of *Arabidopsis thaliana*. *Genome Biol.* **22**, 1–26 (2021).

23. Finally, selection tests shall be spelled out and justified as conservative indirect estimates based e.g. on literature (the very limited neutral loci apparent on Fig.4a suggests non-conservative estimates). Generally, figures and their legends should be better linked to the core text and made comprehensive.

Response: We browsed recent literature (2021 onward) to emphasize that our estimates of selection are very conservative: PCadapt q-values below 0.01 are stringent as most studies use a threshold of 0.05, which is already considered stringent (e.g. Andrews et al. 2021; Gamboa et al. 2021; Giglio et al. 2021; Postolache et al. 2021). For bayescan, we used frequently applied parameters, including a FDR of 0.05% (see also e.g. Reich et al. 2021; Agudelo et al. 2022; Garrido et al. 2021; Vera et al. 2021). Fig. 4a has been misinterpreted, and presents the Mahalanobis distance reflecting the strength of selection, so it is normal that neutral SNPs have very low Mahalanobis values. We added the proportions of natural selection in the caption of Fig. 4a to clarify that in fact the majority of SNPs (>80%) behave neutrally. We finally expanded figure captions to clarify their link with the results.

New L537-548: “**Fig.3. Relationships between heterozygosity and TE characteristics.** Heterozygosity in inbred individuals (genome-wide $F_{is} > 0.6$) was generally low but was elevated near Copia, Harbinger and LINE elements (A). While Copia and Harbinger elements can maintain heterozygosity particularly under inbreeding, only LINE elements are also associated with increased heterozygosity in outcrossing populations (genome-wide $F_{is} < 0.6$) (B). We therefore considered the Copia and Harbinger elements to contribute to the “TE Effect” (box 1, Table 1). As the CACTA superfamily was associated with intermediate levels of heterozygosity in the American *A. lyrata* lineage, we used CACTA as the reference group (intercept) for statistical comparisons with other TE superfamilies in the mixed model output of Table S6. Heterozygosity also depended upon the signature of selection (C and D), with balancing selection typically increasing heterozygosity, particularly downstream of TEs. Background heterozygosity (H_0) refers to heterozygosity near all TE superfamilies but Copia and Harbinger (see Box 1). Significance of factor level estimates are provided in Table S6.”

New L549-558: “**Fig.4. Genome-wide signatures of selection in *A. lyrata* and their association with TE superfamilies.** The Mahalanobis distance represents the deviation from neutrality (A), with 4.86% and 12.17% of genome-wide SNPs manifesting signatures of balancing and divergent selection, respectively (Table S1). Genome-wide signatures of divergent and balancing selection are similar upstream and downstream of TEs (B), but vary considerably among TE superfamilies (C). As compared to background heterozygosity (H_0), Copia and Harbinger elements are associated with elevated levels of nearby heterozygosity under inbreeding (H_{TE}) for SNPs under balancing selection (D and E). While signatures of selection are similar upstream and downstream of TEs, SNPs downstream of TEs are consistently implicated in biological processes related to stress (F). Significance of factor level estimates are provided in Table S6.”

New L559-566: “**Fig.5. Comparison of TE-associated evolution between North-American and European sampling area.** The proportion of outliers associated with TE superfamilies correlates between the two sampling areas (A), even though few SNPs shared signatures of selection between the two sampling areas (B). H_0 (background heterozygosity) is not significantly different from H_{TE} (heterozygosity associated with Copia and Harbinger elements) across outcrossing European *A. lyrata*

genomes (C, Table S8), confirming that the effect of Copia and Harbinger elements on nearby heterozygosity is limited to inbred populations (see also Fig. S3). Blue and red colours represent signatures of balancing and divergent selection, respectively (A,B).

References:

- Andrews, K. R. *et al.* Genomic signatures of divergent selection are associated with social behaviour for spinner dolphin ecotypes. *Mol. Ecol.* **30**, 1993–2008 (2021).
- Gamboa, M. P., Ghalambor, C. K., Scott Sillett, T., Morrison, S. A. & Chris Funk, W. Adaptive divergence in bill morphology and other thermoregulatory traits is facilitated by restricted gene flow in song sparrows on the California Channel Islands. *Mol. Ecol.* **31**, 603–619 (2022).
- Giglio, R. M., Rocke, T. E., Osorio, J. E. & Latch, E. K. Characterizing patterns of genomic variation in the threatened Utah prairie dog: Implications for conservation and management. *Evol. Appl.* **14**, 1036–1051 (2021).
- Postolache, D. *et al.* Genetic signatures of divergent selection in European beech (*Fagus sylvatica* L.) are associated with the variation in temperature and precipitation across its distribution range. *Mol. Ecol.* **30**, 5029–5047 (2021).
- Reich, H. G. *et al.* Genomic variation of an endosymbiotic dinoflagellate (*Symbiodinium 'fitti'*) among closely related coral hosts. *Mol. Ecol.* **30**, 3500–3514 (2021).
- Agudelo, J. F. G. *et al.* Genomic selection signatures in farmed *Colossoma macropomum* from tropical and subtropical regions in South America. *Evol. Appl.* **15**, 679–693 (2022).
- Llanos-Garrido, A., Briega-Álvarez, A., Pérez-Tris, J. & Díaz, J. A. Environmental association modelling with loci under divergent selection predicts the distribution range of a lizard. *Mol. Ecol.* **30**, 3856–3868 (2021).
- Vera, M. *et al.* Genomic survey of edible cockle (*Cerastoderma edule*) in the Northeast Atlantic: A baseline for sustainable management of its wild resources. *Evol. Appl.* **15**, 262–285 (2021).

Reviewers' Comments:

Reviewer #1:

Remarks to the Author:

I reviewed the previous version of this interesting manuscript, and was happy to receive the revision. First, I want to compliment the authors on their very thorough rebuttal and revision, and the way they have taken my feedback (and that of the other reviewers) into consideration. I think the manuscript has improved a lot in terms of clarity and is now a much easier read.

I think there is only one major remaining issue.

The authors have attempted to explain better why they interpret the patterns in their data as signatures of balancing rather than purifying selection. However, I'm still confused.

In the methods for example, the explanation makes sense until about L392. Then they lose me.

L393-395: "Specifically, we considered SNPs with signatures of balancing selection as candidates of purifying selection if their minimum allele frequency (MAF) was (i) smaller than the MAF of neutral SNPs, and (ii) smaller for non-synonymous as compared to synonymous codon usage."

To me, this wording implies that the authors consider negative alpha values as signatures of balancing selection, unless proven otherwise. This seems weird to me, as purifying selection should generally be more frequent than balancing selection. It is not until the discussion that there is some explanation (Fijarczyk & Wiesław Babik 2015).

Although it would be good to incorporate that explanation also in the methods and wherever else appropriate, I would have much less problems with the logic, if it were phrased as follows:

"Specifically, we considered SNPs with signatures of selection (negative alpha values in Bayescan analysis) as candidates of purifying selection if their minimum allele frequency (MAF) was (i) smaller than the MAF of neutral SNPs, and (ii) smaller for non-synonymous as compared to synonymous codon usage. In any other cases, we considered SNPs with negative alpha values as candidates to be under balancing selection."

Maybe you could also provide numbers (in the results) on the number/proportion of SNPs with negative alpha that were considered as under purifying vs. balancing selection. The discussion could then also discuss if the number of SNPs under putative balancing selection is normal/higher than expected, and why this may be.

Related to this, in my opinion, the discussion should more carefully consider what may have caused this signature of balancing selection. The Fijarczyk & Wiesław Babik 2015 explanation is good, but what about other potential explanations? Heterozygote advantage (overdominance)? Or could it also be pseudo-overdominance?? I think so, but pseudo-overdominance seems to have been ignored by the authors. Could it be "real" balancing selection, by the way, through direct selection for heterozygotes??

Line specific comments:

L106 –"heterologous" maybe instead of "nonhomologous"?

L168/169/170 – "can be identified as low frequency signatures of balancing selection". I find this wording confusing. It seems to reflect that authors consider balancing selection the norm, and that purifying selection is a special case of this. However, I do not agree with that view.

I think that SNP-variants with a relatively low frequency are indicative of negative (aka purifying) selection. Why call this "low frequency signatures of balancing selection"? Why is this not simply a "signature of purifying selection"?

L191 – it would be helpful to briefly repeat the crux of the hypothesis.

L218 – authors correctly highlight that their evidence for the importance of TEs for evolution are INDIRECT. The abstract, however, lacks this disclaimer, which I find somewhat misleading.

L232-235 and throughout the ms – the term vital and essential are used in a confusing way (to me), mainly because vital (colloquially) is often used interchangeably with essential, and does not mean the same as vital in a strict genetic sense. It would help if the authors define what they mean by “vital” and perhaps replace with “non-essential”. L233 mentions “signatures” of balancing selection in essential genes, but should this not be purifying?? How would one get balancing selection in essential genes that are supposed to stay intact?? Does the Fijarczyk & Wiesław Babik 2015 explanation apply here`?

L245 “Fitness-related genomic regions” – I have trouble understanding what is meant by this. Is it regions with many non-essential/“vital” genes

L246 – I would say “findings may help explain earlier studies...”

L312-313 – This explanation for why one would end up with signatures of balancing selection (rather than purifying selection) makes sense. I think the authors may want to bring this logic forward, rather than hiding it deep down in the middle of the discussion. This would have taken away much of my confusion.

Fig 1 – the arrow is confusing and seems to have no function. I suggest removing it (or explain what it signifies in the caption).

Maybe also change “Het SNP involved in evolution” to “Het SNP providing evolutionary potential”

Fig 2 – please properly align the different panels.

Fig 4 – In the caption of this figure (and maybe also other figures) it would be good to briefly mention how selection was detected (Bayescan) and how purifying was distinguished from balancing selection.

Reviewer #2:

Remarks to the Author:

I thank the authors for having nicely prepared their answers to reviewers. Most points are clarified.

There is nevertheless one point on which I disagree (point 16):

"Recent findings in *A. thaliana* confirm that non-reference TE insertions appear at low frequency due to their deleterious nature": I think this statement is too general, in my opinion most TE insertions are neutral (except for the families with a tendency to jump into promoter/genic regions). This sentence refers to the work by Baduel et al, thus I encourage the authors to discuss this interpretation with Baduel's co-authors.

My opinion on TE insertions does not affect the conclusions of the present work, and I believe it will be of great interest to the Nature Comm readers.

Reviewer #1

I reviewed the previous version of this interesting manuscript, and was happy to receive the revision. **First, I want to compliment the authors on their very thorough rebuttal and revision**, and the way they have taken my feedback (and that of the other reviewers) into consideration. I think the manuscript has improved a lot in terms of clarity and is now a much easier read.

I think there is only **one major remaining issue**.

1. The authors have attempted to explain better why they interpret the patterns in their data as signatures of balancing rather than purifying selection. However, I'm still confused. In the methods for example, the explanation makes sense until about L392. Then they lose me. L393-395: "Specifically, we considered SNPs with signatures of balancing selection as candidates of purifying selection if their minimum allele frequency (MAF) was (i) smaller than the MAF of neutral SNPs, and (ii) smaller for non-synonymous as compared to synonymous codon usage." To me, this wording implies that the authors consider negative alpha values as signatures of balancing selection, unless proven otherwise. This seems weird to me, as purifying selection should generally be more frequent than balancing selection. It is not until the discussion that there is some explanation (Fijarczyk & Wiesław Babik 2015). Although it would be good to incorporate that explanation also in the methods and wherever else appropriate, **I would have much less problems with the logic, if it were phrased as follows: "Specifically, we considered SNPs with signatures of selection (negative alpha values in Bayescan analysis) as candidates of purifying selection if their minimum allele frequency (MAF) was (i) smaller than the MAF of neutral SNPs, and (ii) smaller for non-synonymous as compared to synonymous codon usage. In any other cases, we considered SNPs with negative alpha values as candidates to be under balancing selection."** Maybe you could also provide numbers (in the results) on the number/proportion of SNPs with negative alpha that were considered as under purifying vs. balancing selection. The discussion could then also discuss if the number of SNPs under putative balancing selection is normal/higher than expected, and why this may be. Related to this, in my opinion, the discussion should more carefully consider what may have caused this signature of balancing selection. The Fijarczyk & Wiesław Babik 2015 explanation is good, but what about other potential explanations? Heterozygote advantage (overdominance)? Or could it also be pseudo-overdominance?? I think so, but pseudo-overdominance seems to have been ignored by the authors. Could it be "real" balancing selection, by the way, through direct selection for heterozygotes??

Response: The logic for defining candidates of purifying selection proposed by the referee reflects our intended definition, hence we have adjusted the corresponding methods section as suggested. We now also provide the number of SNPs with negative alpha considered as purifying (N=1) vs. balancing (N=37) in the caption of a new Supplementary figure, in which we show densities of SNPs under purifying vs. balancing selection (**Fig. S3, panel C**). Pseudo-overdominance is generated by multiple deleterious loci linked in repulsion, consequently generating apparent signatures of balancing selection masking signatures of purifying selection (i.e. low maf particularly at non-synonymous sites). Such signatures are automatically captured by our analysis of purifying selection. Another potential mechanism causing apparent signatures of balancing selection is associative overdominance, where purifying selection can cause apparent heterozygote advantage at linked neutral loci. This type of selection is expected to result in low maf both at non-synonymous and synonymous mutations, but is also clearly underrepresented in our dataset (**Fig. S3, panel C**). In general, our results suggest that TEs are involved in frequency-dependent balancing selection

(aligning with the enrichment signaling and disease related processes), and in selection for heterozygotes (increased heterozygosity in housekeeping genes that may be sensitive to inbreeding depression). Please find below some additions to the discussion accounting for potential other explanations for our observed signatures of balancing selection.

New L410-412 (Methods): *“Specifically, we considered SNPs with signatures of selection (negative alpha values in Bayescan analysis) as candidates of purifying selection if their minimum allele frequency (MAF) was (i) smaller than the MAF of neutral SNPs, and (ii) smaller for non-synonymous as compared to synonymous codon usage. In any other cases, we considered SNPs with negative alpha values as candidates of balancing selection.”*

New L413-414 (Discussion): *“In general, TEs appear to be involved in heterozygote advantage that becomes particularly apparent under inbreeding, and in frequency-dependent balancing selection through their overrepresentation in gene ontology processes related to signaling and (a)biotic stress responses.”*

New L250-252 (Discussion): *“Although experimental validation is required to confirm this heterozygosity “enhancer” activity observed in our study, the heterozygote advantage expected for fitness-related genomic regions that may be sensitive to inbreeding depression is corroborated by the significant role of balancing selection in maintaining heterozygosity under inbreeding (Fig.4E).”*

New L322 (Discussion): *“..., with the enrichment of processes related to signaling and biotic responses particularly supporting a role for frequency-dependent balancing selection (Table S8).”*

New L325-336 (Discussion): *“Note that pseudo-overdominance (at deleterious loci; e.g. Waller 2021) and associative overdominance (at neutral loci; e.g. Gilbert et al. 2020)), both arising from purifying selection at linked recessive deleterious loci, may also contribute to genome-wide signature of balancing selection. Both types of selection are covered by our analysis of purifying selection (see Fig. S3), with pseudo-overdominance and associative overdominance generating signatures of purifying selection (negative alpha values, low maf, predominant non-synonymous codon usage) and apparent purifying selection (negative alpha values, low maf, but variable codon usage), respectively. Although fully disentangling the various types of overdominance remains challenging and some of our “real” signatures of balancing selection may actually reflect pseudo-overdominance (e.g. Waller 2021), our findings suggest that pseudo-overdominance and associative overdominance are strongly underrepresented in our dataset, particularly for the American lineage (Fig. S3).”*

New Fig. S3. “Minimum allele frequency (MAF) distributions. Panels A and B present non-synonymous vs. synonymous SNPs, for the American (A) and European (B) *A. lyrata* lineage. MAF means are presented with standard errors per TE superfamily. TE superfamilies with too low sample sizes ($N=1$) are transparent (e.g. LAR in both panels). TE superfamilies associated with low frequency signatures of balancing selection are considered as candidates of purifying selection if their MAF is smaller than expected under neutrality, i.e. smaller than MAF of neutral SNPs, and smaller for non-synonymous than for synonymous codon usage. We find that neutral SNPs have the same MAF distribution in both lineages. For the American lineage, we further find that all TE superfamilies with $N>1$ are characterized by high frequency signatures of selection, indicating little purifying selection. For the European lineage, however, we identify signatures of purifying selection in several TE superfamilies (e.g. MuDR and Cacta), where MAF was smaller for SNPs with signatures of balancing selection than for neutral SNPs, and where non-synonymous SNPs in particular were associated with low MAF. **Panels C and D represent MAF density distributions across all non-synonymous (green) and synonymous (purple) SNPs, with signatures of balancing and purifying selection (C) vs. neutral SNPs (D). The orange rectangle in panel C represents the 95% CI of the mean maf of neutral SNPs (9.02-10.64); therefore, all SNPs in panel C left from this orange rectangle are candidates for purifying selection (non-synonymous SNPs with low maf; $N=1$) and/or for associative overdominance (non-synonymous and synonymous SNPs; $N=1 + 0$). A total of 37 SNPs (97.4% out of all SNPs with $\alpha < 0$) are candidates for real balancing selection arising from heterozygote advantage or frequency-dependent selection.”**

New references:

- Gilbert KJ, Pouyet F, Excoffier L, Peischl S. 2020. Transition from background selection to associative overdominance promotes diversity in regions of low recombination. *Current Biology* 30: 101-107.
- Waller DM. 2021. Addressing Darwin's dilemma: Can pseudo-overdominance explain persistent inbreeding depression and load? *Evolution* 75: 779-793.

Line specific comments:

2. L106 –“heterologous” maybe instead of “nonhomologous”? → Done (New L106)

3. L168/169/170 – “can be identified as low frequency signatures of balancing selection”. I find this wording confusing. It seems to reflect that authors consider balancing selection the norm, and that purifying selection is a special case of this. However, I do not agree with that view. I think that SNP-variants with a relatively low frequency are indicative of negative (aka purifying) selection. Why call this “low frequency signatures of balancing selection”?? Why is this not simply a “signature of purifying selection”?

Response: We fully agree and rephrased accordingly.

New L169-71: “While SNPs near CACTA elements appeared to be associated with low minor allele frequencies, which could point to purifying selection acting on deleterious alleles, we found signatures of purifying selection to be generally rare”.

4. L191 – it would be helpful to briefly repeat the crux of the hypothesis.

New L191: “In line with our final hypothesis that the role of TEs in evolution is consistent across sampling areas (Fig.1: Q3), we finally found that ...”

5. L218 – authors correctly highlight that their evidence for the importance of TEs for evolution are INDIRECT. The abstract, however, lacks this disclaimer, which I find somewhat misleading.

Response: we toned down the abstract a little.

New L24: Together, our study provides a novel explanation for the success of self-fertilizing species.” → “Together, our study provides a novel hypothesis for the success of self-fertilizing species.”

6. L232-235 and throughout the ms – the term vital and essential are used in a confusing way (to me), mainly because vital (colloquially) is often used interchangeably with essential, and does not mean the same as vital in a strict genetic sense. It would help if the authors define what they mean by “vital” and perhaps replace with “non-essential”. L233 mentions “signatures” of balancing selection in essential genes, but should this not be purifying?? How would one get balancing selection in essential genes that are supposed to stay intact?? Does the Fijarczyk & Wiesław Babik 2015 explanation apply here`?

Response: We absolutely agree that mutations arising in essential genes have a high potential of being deleterious and purged away, but the rare ones that are not purged can be neutral or even beneficial to the functioning of these essential genes. We link signatures of balancing selection to processes that become deleterious when their underlying genetic basis becomes homozygous (cfr. inbreeding depression). We thus allow essential genes to be heterozygous (which does not necessarily results in malfunctioning and purifying selection, in line with our findings of limited purifying selection, please see comment **1**). In that sense, “vital” is a synonym for “essential”. Yet we understand that this is easy to confuse with how one typically defines purifying selection. We thus clarified as requested, and we are open to additional suggestions for improvement if that is recommended by the referee.

New L237-240: “Importantly, while mutations arising in essential genes that are vital to organismal functioning are typically considered harmful and the target of purifying selection, our results suggest that mutations occasionally persist in essential genes where they can contribute to balancing evolution.”

7. L245 “Fitness-related genomic regions” – I have trouble understanding what is meant by this. Is it regions with many non-essential/“vital” genes?

Response: We refer to genomic regions that directly impact fitness or organismal functioning, i.e. vital genomic regions.

New L245: “... in fitness-related genomic regions containing vital genes directly impacting fitness or organismal functioning.”

8. L246 – I would say “findings may help explain earlier studies...” → Done (**New L253**)

9. L312-313 – This explanation for why one would end up with signatures of balancing selection (rather than purifying selection) makes sense. I think the authors may want to bring this logic forward, rather than hiding it deep down in the middle of the discussion. This would have taken away much of my confusion.

Response: We fully agree and introduced this explanation in the first paragraph of the discussion.

New L218-220: “In general, TEs appear to be involved in heterozygote advantage that becomes particularly apparent under inbreeding, and in frequency-dependent balancing selection through their overrepresentation in gene ontology processes related to signaling and (a)biotic stress responses.”

10. Fig 1 – the arrow is confusing and seems to have no function. I suggest removing it (or explain what it signifies in the caption). Maybe also change “Het SNP involved in evolution” to “Het SNP providing evolutionary potential”.

Response: We now refer to “SNPs providing evolutionary potential” in Fig. 1, and clarified the arrow in the caption.

New L543 (Fig. 1): “The arrow in the lower panel (Q3) hypothesizes extrapolation of the findings observed for the American lineage towards the European lineage.”

11. Fig 2 – please properly align the different panels. → Done

12. Fig 4 – In the caption of this figure (and maybe also other figures) it would be good to briefly mention how selection was detected (Bayescan) and how purifying was distinguished from balancing selection.

New L567-569 (caption Fig. 4): “Balancing selection is based on negative alpha values in Bayescan analysis (loci with significantly reduced genetic differentiation), and is distinguished from signatures of purifying selection because they generally represent high-frequency variants (see Fig. S3).”

New L577-579 (caption Fig. 5): “Balancing selection is based on negative alpha values in Bayescan analysis (loci with significantly reduced genetic differentiation), and is distinguished from signatures of purifying selection because they generally represent high-frequency variants (see Fig. S3).”

Reviewer #2:

I thank the authors for having nicely prepared their answers to reviewers. Most points are clarified. There is nevertheless one point on which I disagree (point 16): "Recent findings in *A. thaliana* confirm that non-reference TE insertions appear at low frequency due to their deleterious nature": I think this statement is too general, in my opinion most TE insertions are neutral (except for the families with a tendency to jump into promoter/genic regions). This sentence refers to the work by Baduel et al, thus I **encourage the authors to discuss this interpretation with Baduel's co-authors**. My opinion on TE insertions does not affect the conclusions of the present work, and I believe it will be of **great interest to the Nature Comm readers**.

Response: We thank the referee for his/her previous efforts in the review process, and for appreciating the value of our work. As suggested, we contacted Dr. Pierre Baduel (pierre.baduel@ens.psl.eu) for elaborating on the issue regarding non-reference TE insertions. He interprets our statement as correct and fully in line with the results from his and others research. Please find below his expert opinion on our statement "Recent findings in *A. thaliana* confirm that non-reference TE insertions appear at low frequency due to their deleterious nature".

Dr. Pierre Baduel: "Your statement seems perfectly in line with our interpretation and we have several lines of evidence to support it. Indeed, we found the proportion of low-frequency TIPS to be higher than of non-synonymous SNPs, and most of these low-frequency TIPS are actually genic or near genic. This is in line with the insertion preferences described for several TE families in the near absence of selection in transposition-accumulation lines (Quadrana et al. Nat Comm. 2019). Finally, these genic TE insertions are associated with strong effects on the expression they are located within and are almost completely purged from the population at high-frequency. Together these observations strongly support that non-reference TIPS segregate at low-frequency due to their mostly deleterious nature. Similar excesses of low-frequency of TE insertion polymorphisms are almost systematically found in other species as well (*Drosophila*, *Brachypodium*...)."

Reviewers' Comments:

Reviewer #1:

Remarks to the Author:

With a super-clear rebuttal and final edits, the authors dealt with all remaining comments and I have no further comments. I look forward to see this paper published!

Typo in L328: signatureS

Reviewer #2:

Remarks to the Author:

I thank the authors for taking the time to contact Dr Baduel. His statement is clearly exposed. This was one of my most interesting and constructive reviewing process. Many thanks for this exchange.

Reviewer #1:

With a super-clear rebuttal and final edits, the authors dealt with all remaining comments and I have no further comments. I look forward to see this paper published!

1. Typo in L328: signatureS → Corrected

Reviewer #2:

I thank the authors for taking the time to contact Dr Baduel. His statement is clearly exposed. This was one of my most interesting and constructive reviewing process. Many thanks for this exchange.